# Characterization of the development of the mouse cochlear epithelium at the single cell level

Likhitha Kolla[1,6], Michael C. Kelly[1,6], Zoe F. Mann [2], Alejandro Anaya-Rocha[1], Kathryn Ellis[1], Abigail Lemons[1], Adam T. Palermo [3], Kathy S. So[3], Joseph C. Mays [1], Joshua Orvis[4], Joseph C. Burns[3], Ronna Hertzano[4,5], Elizabeth C. Driver [1] & Matthew W. Kelley [1✉]

Mammalian hearing requires the development of the organ of Corti, a sensory epithelium comprising unique cell types. The limited number of each of these cell types, combined with their close proximity, has prevented characterization of individual cell types and/or their developmental progression. To examine cochlear development more closely, we transcriptionally profile approximately 30,000 isolated mouse cochlear cells collected at four developmental time points. Here we report on the analysis of those cells including the identification of both known and unknown cell types. Trajectory analysis for OHCs indicates four phases of gene expression while fate mapping of progenitor cells suggests that OHCs and their surrounding supporting cells arise from a distinct (lateral) progenitor pool. Tgfβr1 is identified as being expressed in lateral progenitor cells and a Tgfβr1 antagonist inhibits OHC development. These results provide insights regarding cochlear development and demonstrate the potential value and application of this data set.

[1] Laboratory of Cochlear Development, National Institute on Deafness and Other Communication Disorders, National Institutes of Health, Bethesda, MD 20892, USA. [2] Centre for Craniofacial and Regenerative Biology, King's College London, London, UK. [3] Decibel Therapeutics, 1325 Boylston, Str., Suite 500, Boston, MA 02215, USA. [4] Institute for Genome Sciences, University of Maryland School of Medicine, Baltimore, MD 21201, USA. [5] Department of Otorhinolaryngology Head and Neck Surgery, Anatomy and Neurobiology, and Institute for Genome Sciences, University of Maryland School of Medicine, Baltimore, MD 21201, USA. [6] These authors contributed equally: Likhitha Kolla, Michael C. Kelly. ✉email: kelleymt@nidcd.nih.gov

The organ of Corti (OC), located in the floor of the scala media of the cochlea, acts as the primary sensory transducer of sound in mammals. This structure comprises a highly diverse cellular mosaic that includes two different types of mechanosensory hair cells (HCs), and an undefined number of associated supporting cell (SC) types (Fig. 1a). All of these cells are believed to arise from a developmental equivalence group referred to as the prosensory domain[1]. The results of embryologic manipulations and molecular genetic experiments suggest that otocyst precursor cells proceed through several rounds of lineage restriction that progressively specify subsets of cells as prosensory cells, and ultimately as either HCs or SCs (ref. [1]).

To examine the transcriptional changes that occur during the formation of the OC, we dissociate cochlear duct cells at four developmental time points and then capture individual cells for analysis using single-cell RNAseq. Results identify multiple

**Fig. 1 Characterization of cell types in the P1 cochlea. a** Line drawing of a cross section of the floor of the cochlear duct at P1. Distinct cell types within the organ of Corti (OC) are color coded. **b** Heat map for ~14,000 cochlear cells collected from four separate experiments at P1. Top 25 differentially expressed (DE) genes for the 15 identified clusters are shown. Cellular identity for each cluster is indicated by a color bar at the top of the heat map, which corresponds to the legend in **a**, and by a cell name at the bottom. **c** tSNE plot for the same cells as in **b**. Cluster identities are indicated. **d** Violin plots showing normalized log-transformed expression values for the top five DE genes for each cell type (color coded as in **c**) by comparison with all other P1 cells (gray on the right in each graph). Bars indicate median expression level. **e** Upper left panel, tSNE plot of cells determined to be derived from KO (between the OC and medial edge of the cochlear duct). Lower left panel, feature plot for the same cells as in the upper panel indicating high expression of *Otoa*, *Calb1*, and *Fabp7* (based on color) in different clusters of cells. Lower right panel, cross sections through the cochlear duct at P1, illustrating expression of CALB1 in the medial region of KO and FABP7 directly adjacent to the OC (arrow; scale bars, 20 μm). Lowest panel shows high-magnification view of expression of FABP7 (arrow, gray scale) at the lateral KO border (green line; scale bar, 10 μm). Upper right panel, summary diagram of the spatial distribution of KO cell clusters at P1. HC hair cells, IPhC inner phalangeal cells/border cells, IPC inner pillar cells, OPC outer pillar cells, DC1/2 Deiters' cells rows 1 and 2, DC3, Deiters' cells row 3, HeC Hensen's cells, CC/OSC Claudius cells/outer sulcus cells, IdC interdental cells, ISC inner sulcus cells, KO Kölliker's organ cells, L.KO lateral Kölliker's organ cells, M.KO medial Kölliker's organ cells, OC90 OC90+ cells.

unique cell types at each time point, including both known types, such as HCs and SCs, and previously unknown cell types, such as multiple unique cell types in Kölliker's organ (KO). Cells collected from E14 and E16 cochleae include prosensory cells; however, unbiased clustering indicates two distinct populations. Fate mapping of one of these populations demonstrates a strong bias toward lateral fates (OHCs and surrounding support cells), suggesting that these cells represent a unique lateral prosensory population. Differential expression analysis of the lateral prosensory cells identifies multiple genes that are exclusively expressed in this region, including *Tgfβr1* (transforming growth factor β receptor 1) which is mutated in Ehlers–Danlos and Loeys–Dietz syndromes[2,3], both of which can include hearing loss. To examine the role of Tgfβr1, we use an in vitro approach to block Tgfβr1 signaling in developing cochlea. Results indicate an inhibition of OHC development. Overall, these results demonstrate the quality and potential application of this data set to better understand the transcriptional changes that occur during the development of the cochlear duct.

## Results

**Characterization of cochlear cell types at P1**. As a first step, 14,043 cells from the floor of the P1 cochlear duct were isolated and analyzed (Fig. 1b, c, Supplementary Data 1). Unbiased clustering identified 15 distinct groups of cells with discrete patterns of gene expression (Fig. 1b, c). An examination of the top 25 genes defining each cluster was used to assign identities to each group (Fig. 1c, Supplementary Data 1). HCs were identified based on expression of known marker genes, such as *Myo6*, *Myo7a*, *Pvalb*, and *Cib2* (refs. [4–6]; Supplementary Fig. 1). Next, to identify markers for each cell type, gene expression was compared between each cell type and all other cell types (Fig. 1d). These comparisons identified markers for several known cell types, including *Ccer2*, *Acbd7*, *Rprm*, and *Cd164l2* in HCs, *Pmch* in Hensen's cells, *Emid1* and *Npy* in IPCs, and *Matn4* in inner phalangeal cells (Fig. 1d, Supplementary Data 2). DCs could be separated into either first/second or third row with known markers of third row DCs, such as *Lgr5* and *Fgf3* (refs. [7,8]), restricted to that cell population (Supplementary Fig. 1). OPCs and first/second row DCs were transcriptionally similar (Fig. 1b, d), but IPCs were transcriptionally distinct from other SC types (Fig. 1b, c). Finally, a small cluster of cells strongly expressed *Otoconin90* (Fig. 1b, c), which is restricted to the cochlear roof[9]. These cells likely represent cochlear roof cells that were included in the captured samples to ensure the entire medial to lateral cochlear floor was represented. In addition to known cell types within the OC, the P1 data set also includes cells from KO, a transient group of epithelial cells located between the OC and medial side of the cochlear duct[10] (Fig. 1a). KO is an intriguing region of the cochlear duct that has several different functions

during cochlear development. In particular, cells within KO play a role in the development of the tectorial membrane[11], the generation of spontaneous activity required for maturation of spiral ganglion neurons[12] and some cells within this region retain prosensory potential[13–15]. However, since KO cells are morphologically homogenous, the extent of transcriptional heterogeneity was unclear. Our initial analysis identified six clusters of cells that were classified as located between the OC and the medial edge of the duct (interdental cells, inner sulcus cells, and KO1–4). To characterize these cells more thoroughly, they were grouped into a new data set and reanalyzed. A t-distributed Stochastic Neighbor Embedding (tSNE) plot for these cells indicated a linear distribution for the six clusters along the tSNE1 axis (Fig. 1e). The distribution of cell types suggested the possibility that the position along tSNE1 could reflect cellular position along the medial–lateral axis of the cochlear duct (Fig. 1a). To determine if this was the case, expression of cell-type-specific genes was examined. Localization of known markers for interdental cells, such as *Otoa*[16], indicated restricted expression in the cluster located on the left-hand edge of the tSNE plot. In contrast, expression of *Fabp7*, which is restricted to cells located directly adjacent to the OC (ref. [17]; Fig. 1e), was localized to the group of cells at the extreme right-hand edge of the tSNE plot. Similarly, *Calb1*/Calb1, which is expressed in a central region of KO, was localized in the middle of the tSNE plot (Fig. 1e). These results were consistent with the tSNE1 axis reflecting cellular position along the medial–lateral axis of the cochlear duct. For additional confirmation, we identified genes that were expressed in specific cell groups within KO (Supplementary Fig. 2) and then compared their distribution along tSNE1 with published positional data[11,18,19]. Based on these results, the different clusters arising from KO were renamed to reflect their position along the medial–lateral axis (Fig. 1e).

To validate some of the cell-type-specific genes identified in the P1 data set, we used single-molecule fluorescent in situ hybridization (smFISH) to localize transcripts for several different cell types in cross sections from P1 cochleae. *Pvalb*, a known HC marker[20], was used as a positive control (Fig. 2a). Four genes that were HC specific based on scRNAseq results, but had not been reported to be expressed in HCs, *Rprm*, *Cd164l2*, *Ccer2*, and *Gng8*, and one gene that showed restricted expression in IHCs and KO, *Tbx2*, were also examined (Fig. 2b–e). All four genes showed patterns of expression that were consistent with the single cell results although *Tbx2* was also expressed at lower levels in surrounding SCs. Next, expression of two known SC genes, *Sox2* and *Cdkn1b* was compared with a candidate SC gene, *Prss23* that was detected in all SCs, as well as some KO cells (Fig. 1d). This pattern of expression was confirmed by smFISH (Fig. 2i). *Npy* was among the top five differentially expressed (DE) genes in IPCs and showed a high degree of specificity to this cell type

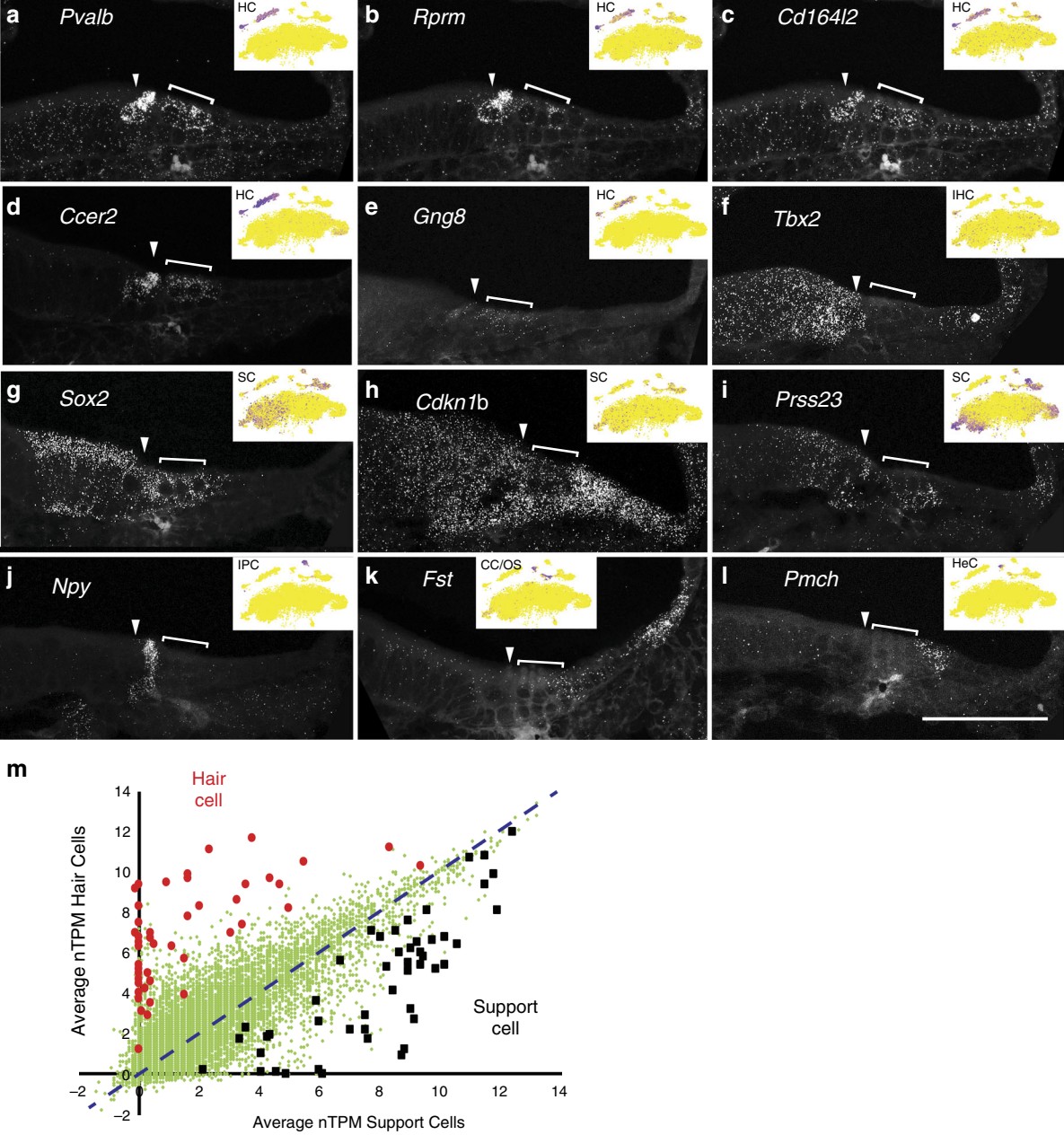

**Fig. 2 Validation of cell-type-specific markers at P1. a–l** In situ hybridization in P1 cochlear cross sections using smFISH. HCs are localized based on expression of *Pvalb*. Inset: feature plot for each gene in the P1 *t*SNE (high expression is indicated in blue). For all panels, the IHC is indicated with an arrow and the three OHCs are indicated by a bracket. HC markers *Rprm* **b**, *Cd164l2* **c**, *Ccer2* **d**, and *Gng8* **e**, are all restricted to HCs. *Tbx2* **f**, which was found only in the IHC cluster, is actually expressed in both IHCs and parts of KO. *Sox2* **g** and *Cdkn1b* **h**, markers of SCs, are not expressed in HCs but are expressed in surrounding cells. *Prss23* **i** shows a similar localization to cells surrounding the HCs. *Npy* **j** is only expressed in pillar cells located between the IHC and the first OHC. *Fst* **k** is localized in cells lateral to the OC and *Pmch* **l** is specific for Hensen's cells located just adjacent to the third OHC. **m** Scatter plot comparing differential gene expression for P1 HCs and lateral SCs (see Methods section) with the results from Burns et al.[4]. Green diamonds indicate average nTPM counts for HCs versus SCs in Burns et al.[4]. Red circles indicate the values for the top 50 DE HC genes, and black squares indicate the values for top 50 DE SC genes from this study. See Supplementary Data 3 for a full list of gene names and values. Note that all P1 DE HC genes were more highly expressed in HC from Burns et al.[4] as well. Six P1 DE HC genes and three P1 DE SC genes were not detected in either cell type (see Methods section for details and Supplementary Data 3 (Col. AC-AK for full list of gene identities)). Scale bar in **l** (same in **a–k**), 50 µM.

(Figs. 1d and 2f). Finally, two candidate outer sulcus cell markers were examined by comparing their expression patterns with *Bmp4* (ref. [21]). Consistent with the single-cell data, *Fst* is expressed in Hensen's cells and a lateral population of outer sulcus cells that abuts the *Bmp4* population[22], while *Pmch* is only expressed in Hensen's cells (Fig. 2k, l).

Next, to compare the accuracy of the cell-type-specific gene expression profiles generated using the 10X Chromium transcript end-counting method with scRNAseq profiling previously performed with full-length transcripts, the gEAR (https://umgear.org/) compare tool was used to generate a scatter plot for average gene expression in P1 cochlear HCs and lateral SCs

from a previous study[4]. The top 50 DE genes in the most comparable cell type groups from this study (Supplementary Data 3, see Methods section for details) were then mapped onto the scatter plot. With the exception of nine genes that were not detected at all in the Burns et al.[4] study, all DE genes were more highly expressed in the predicted cell types (Fig. 2m).

**Outer HC development**. The initial analysis of P1 cochlear cells clustered all HCs into a single group. However, transcriptional differences between HC types are known to be present by P0 (ref. [23]). Therefore, we isolated and reclustered the 1047 P1 HCs. When this data set was analyzed, four cell clusters were identified (Fig. 3a). Based on the gene expression (Fig. 3b), these cells were classified as 187 inner or 860 outer HCs (IHCs and OHCs) at two separate stages of development. These four groups were labeled as immature IHCs and OHCs, or simply IHCs and OHCs. Consistent with their more rapid rate of maturation, a greater number of unique genes were localized in IHCs (Fig. 3b). These included known IHC markers, such as *Fgf8*, *Atp2a3*, *Cabp2*, and *Shtn1* (refs. [24,25]), but also identified *Kcnj13* and *Fam19a3* as candidate markers of IHCs at P1. In contrast, unique outer HC genes were limited to *Calca* (*Cgrp*), *Serpina1c*, *Veph1*, *Cacng2* (*Stargazin*), *Strip2*, and *Msln1*.

HCs within the OC are known to develop in a gradient that extends along the basal-to-apical axis. As a result, the HCs collected at P1 were at different stages of maturity. The distribution of OHCs in the *t*SNE plot showed a roughly linear pattern that we suspected might be reflective of a gradient in maturation. To examine this, the OHCs were ordered along a pseudotime gradient using Monocle[26]. We identified four distinct patterns of gene expression labeled as OHC1–4 (Fig. 3c, Supplementary Data 4). OHC1 included genes, such as *Sox2*, a marker of HC progenitors[27], that were only expressed in cells at the beginning of the trajectory (Fig. 3d). OHC2 included genes, such as *Atoh1* and *Insm1*, transcription factors (TFs) that are transiently expressed in all HCs (*Atoh1*) or only in OHCs (*Insm1*)[23,28]. In contrast, OHC3 and OHC4 included genes, such as *Atp2b2*, a calcium ATPase associated with hearing loss and expressed in stereocilia[29], *Calb2*, a calcium-binding protein that is known to be upregulated as HCs mature, and *Tmc1* (Fig. 3d). These results provide a catalog of genes that are expressed as OHCs progress through different phases of development. To create a more inclusive characterization of OHC development, we next clustered all OHCs from E14, E16, P1, and P7, and used Monocle to generate an additional trajectory. Results indicate a similar developmental pattern separating into four phases (Supplementary Fig. 3, Supplementary Data 5).

Next, we sought to identify gene regulatory networks and corresponding TFs that might regulate the development of different cell types within the P1 data set. SCENIC combines gene co-expression with cis-regulatory motif analysis to compensate for the technical variation and decreased sensitivity present in single-cell data sets[30]. To compensate for low gene expression values in individual cells, randomly sub-setted groups of cells from each identified P1 cluster were averaged prior to analysis (see Methods section). The resulting regulatory *t*SNE identified cell type clusters that were similar to those generated in the transcriptional *t*SNE (Figs. 3e and 1c, Supplementary Data 6). An examination of TF regulons that were highly localized to HCs revealed several TFs that are known, within the inner ear, to be restricted to HCs, including Lhx3, Atoh1, Pou4f3, and Barhl1 (refs. [31–33]). However, several additional TFs, including Brf2 and Usf2, were also identified (Fig. 3e, Supplementary Data 6), a result that is consistent with previous bulk RNAseq experiments demonstrating expression of both TFs in HCs (refs. [34–36]).

Differentially assigned TFs for each P1 cell cluster were also identified (Supplementary Data 6). To examine TFs that might be unique to IHCs or OHCs, regulon analysis was examined within the HC clusters (Fig. 3f). The TF, Bdp1, was assigned to only IHCs, while Arnt was largely restricted to more mature HCs of both types. Neither gene has been localized to HCs, but Bdp1 has been linked with human hereditary hearing loss[37]. SCENIC analysis was also performed for all identified cell types in the E14, E16, and P7 data sets (Supplementary Data 7–9).

**Lateral cochlear cells arise from a restricted progenitor pool**. To examine development of the lateral compartment of the OC prior to OHC differentiation, single cells were collected from the floor of the duct at E14 (4495 total) and E16 (7961 total; Supplementary Data 1). Unbiased clustering was performed to identify discrete cell groups, (Fig. 4a, b) and DE genes for each cluster were identified (Supplementary Figs. 4 and 5, Supplementary Data 1). At E14, a limited number of HCs were identified based on expression of early HC genes, such as *Atoh1*. At E16, in addition to IHCs and OHCs, several other clusters of cells could be identified, including inner phalangeal cells, Hensen's cells, and IPCs (Fig. 4b, Supplementary Data 2).

To identify the population of prosensory cells in the E14 data set, expression of two known prosensory markers, *Cdkn1b* and *Sox2*, was examined. Two clusters contained cells that were positive for *Cdkn1b*, *Sox2*, or both (Fig. 4c). To determine whether these cells might represent separate clusters of medial and lateral prosensory cells (MPsCs and LPsCs), expression of *Fgf20*, a medial prosensory marker[38], and *Fgfr3* and *Prox1*, lateral prosensory markers[39,40], were examined. Consistent with this hypothesis, *Fgf20* and *Fgfr3/Prox1* segregated into separate prosensory clusters (Fig. 4c, d, Supplementary Fig. 6). To determine whether these clusters represent discrete prosensory populations, violin plots for *Fgf20*, *Fgfr3*, and *Prox1* were generated for the E14 and E16 data sets. *Fgf20* was expressed in MPsCs and IHCs, while *Fgfr3* and *Prox1* were expressed in LPsCs, IPCs, and OHCs (Supplementary Fig. 7). While the expression of *Fgf20* or *Fgfr3* in differentiating cell types that do not express these genes in their mature state, such as IHCs and OHCs, suggested that these cells represented a transitional phase, we wanted to confirm that distinct MPsC and LPsC populations exist as early as E14. Therefore, we performed fate mapping using *Fgfr3icre*; *R26RtdTomato* mice that were induced on E13.5, E14.5, or E15.5, and then analyzed at P0. Results indicate that 98.3% of *Fgfr3+*-progenitors at E14–E16 go on to develop as OHCs, DCs, OPCs, or IPCs (Fig. 4e). This finding was consistent with previous work demonstrating that *Fgfr3* expression is restricted to OHCs, DCs, and PCs by P1 (refs. [39,41]). While lineage tracing does not prove that the fates of *Fgfr3+*-prosensory cells are restricted at the E13.5–E15.5 time points, they do demonstrate that in the absence of developmental perturbations, the early *Fgfr3+* population of cells gives rise almost entirely to cell types located within the lateral domain.

Based on these results, we clustered MPsCs, IHCs, and IPhCs from E14, E16, and P1, and preformed trajectory analysis using Monocle, which generated a single bifurcation that split MPsCs into either IPhCs or IHCs (Fig. 4f, Supplementary Fig. 8). Transcriptional changes along the two branches are illustrated in Supplementary Fig. 8 and Supplementary Data 10. In contrast with medial cell types, Monocle analysis of LPsCs, OHCs, and DCs failed to produce a bifurcated trajectory (Fig. 4f). Instead cells formed a continuum with prosensory cells located in the center and extending, in general, toward OHCs on one end and DCs on the other end. The reasons for the lack of a bifurcation are unclear but could suggest that LPsCs collected at E14 already

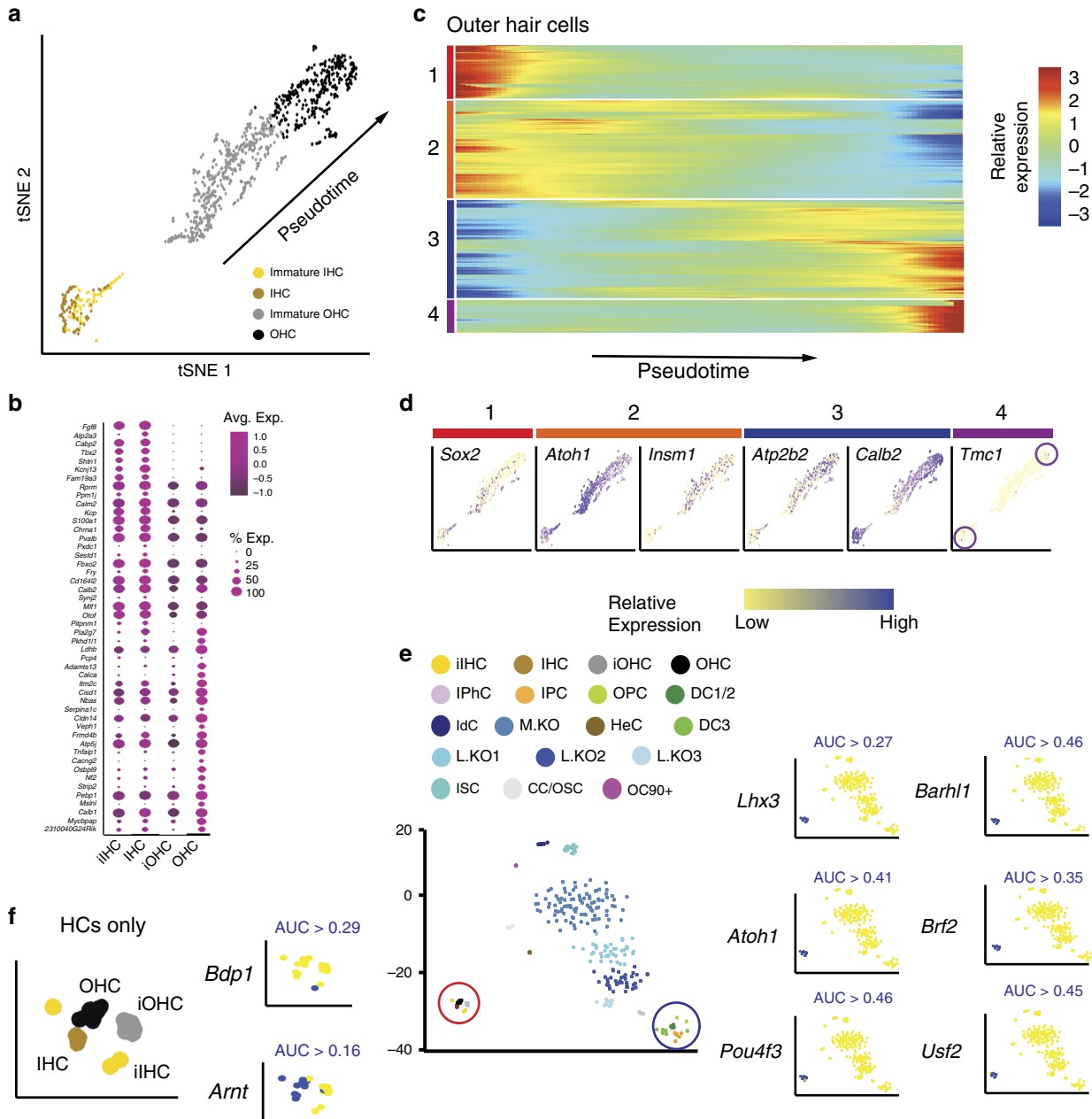

**Fig. 3 Development of OHCs. a** *t*SNE plot for HCs isolated at P1. Both IHCs and OHCs can be clustered into immature and more mature cell groups based on gene expression. Arrow indicates trajectory of pseudotime (see **c**, **d**). **b** Dot plot showing differentially expressed genes between IHCs and OHCs. IHCs exclusively express several genes, including *Fgf8*, *Atp2a3*, *Cabp2*, *Tbx2*, and *Shtn1*. In contrast, a more limited number of genes are exclusively expressed in OHCs, including *Calca*, *Serpina1c*, *Veph1*, *Cacng2*, *Strip2*, and *Msln1*. This may reflect the delay in OHC maturation relative to IHC. **c** The P1 OHCs were analyzed using Monocle to generate a pseudotime trajectory. Analysis of gene expression profiles along pseudotime identified four different developmental phases. **d** Feature plots for a representative gene or genes from each of the four OHC developmental phases. Expression of *Tmc1* was limited to the most mature IHCs and OHCs (purple circles) **e** Left-hand panel, *t*SNE plot based on identification of transcriptional regulons using SCENIC. The HC and SC clusters are indicated by red (HC) and blue (SC) circles. Right-hand panel, feature plots for representative HC-specific regulons (blue indicates high expression). Lhx3, Atoh1, Pou4f3, and Barhl1 are known HC-specific transcription factors (TFs). Brf2 and Usf2 have not been previously localized to hair cells but are present in HCs from previous bulk RNAseq experiments (see text). **f** Left-hand panel, expanded regulon *t*SNE for the HCs. Right-hand panel, two additional TFs, Bdp1 and Arnt1 are localized to HC subsets. HC hair cells, IPhC inner phalangeal cells/border cells, IPC inner pillar cells, OPC outer pillar cells, DC1/2 Deiters' cells rows 1 and 2, DC3, Deiters' cells row 3, HeC Hensen's cells, CC/OSC Claudius cells/outer sulcus cells, IdC interdental cells, ISC inner sulcus cells, KO Kölliker's organ cells, L.KO lateral Kölliker's organ cells, M.KO medial Kölliker's organ cells, OC90 OC90+ cells.

express genes that are specific for either a DC or OHC fate. Finally, we generated a trajectory analysis using all prosensory cells, HCs, DCs, and IPhCs (Supplementary Fig. 9). The results appeared very similar to the results for the lateral cell type trajectory. These results suggest that collecting additional prosensory cells at E13 and/or E12 might provide a more comprehensive trajectory.

**Activation of Tgfβr1 is required for OHC development.** The demonstration of unique gene expression and restricted cell fate

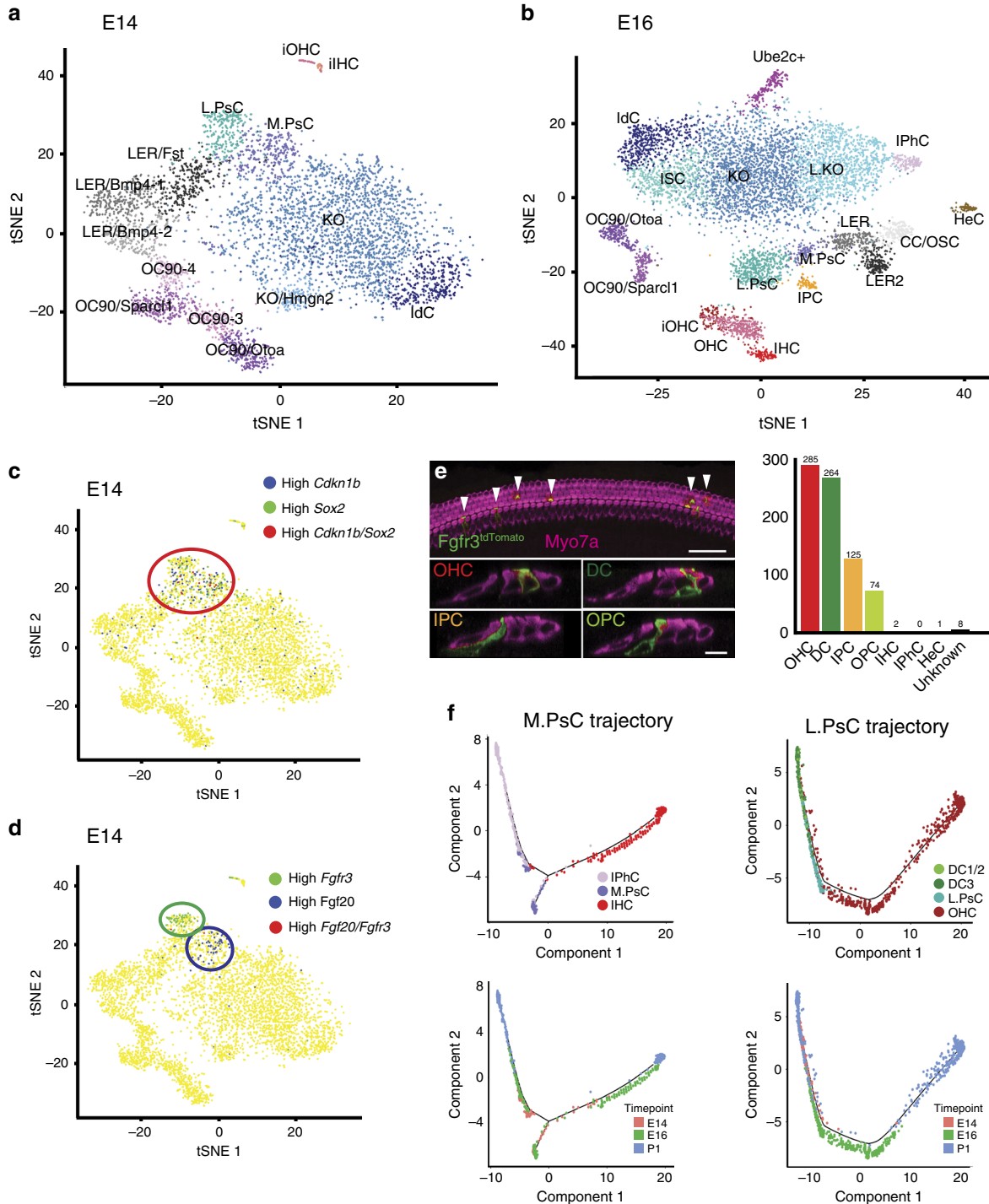

within the lateral prosensory domain strongly suggests that cells located within this region differ from MPsCs. To identify factors that might play a role in the development of LPsCs, gene expression in this group was compared against all other E14 cells. The resulting gene list was then screened to identify those genes that were exclusively expressed by LPsCs (Supplementary Fig. 10, Supplementary Data 11). This list contained genes that were known to be expressed in LPsCs, including *Prox1*, *Bmp2*, *Ngfr*, and *Nrcam*[39,40,42,43], and additional markers, including *Tgfβr1*, *Fzd9* (Frizzled9), *Elmo1*, and *Lsamp* (Fig. 5a, Supplementary Fig. 10). *Tgfβr1* and *Fzd9* are particularly notable as mutations in *Fzd9* have been implicated in Williams syndrome[44], which includes hearing loss[45], while mutations in *Tgfβr1* have been

implicated in Ehlers–Danlos and Loeys–Dietz syndromes, both of which can include hearing loss[2,3,46]. Feature plots for expression of *Tgfβr1* and *Fzd9* in the E14 and E16 data sets confirmed concentrated expression of both genes in LPsCs, although sporadic expression in other cells types was also observed (Fig. 5b, Supplementary Fig. 10). To confirm that these genes are expressed in LPsCs, we first examined their expression in cochlear cross sections at E14.5 and E15.5 at the EurExpress (www.eurexpress. org) and Allen Brain Atlas (developingmouse.brain-map.org) web sites. Each gene was expressed in a restricted band of cells within the cochlear duct (Supplementary Fig. 10). To more precisely localize the expression of both *Tgfβr1* and *Fzd9*, we performed smFISH on cochlear sections from E16 and P1 (Fig. 5c). Strong

**Fig. 4 Medial and lateral prosensory cells are transcriptionally distinct by E14. a, b** tSNE plots for cochlear cells isolated from three separate experiments at E14 **a** and E16 **b**. Cluster identities are based on analysis of gene expression. **c** Feature plot illustrating expression of *Cdkn1b* (blue dots represent cells with high expression of *Cdkn1b*), *Sox2* (green dots; high expression of *Sox2*), or high expression of both (red dots) in single cells isolated at E14. The red circle highlights concentrated expression. **d** Feature plot illustrating mutually exclusive expression of *Fgfr3* (high expressing cells are in green) and *Fgf20* (high expressing cells are in blue) in the same region as in **c**. The green and blue circles indicate concentrated expression of *Fgfr3* (green) and *Fgf20* (blue) **e** Left-hand panel, fate mapping of *Fgfr3*+ cells labeled between E14 and E16. Examples of cells that have developed as OHC, DC, IPC, and OPC are illustrated. Right-hand panel, histogram of fates of *Fgfr3*+-cells. Numbers indicate total number of cells for each cell type. Over 98% of all *Fgfr3*+ cells develop as cells within the lateral domain of the OC. Source data are supplied as a Source data file. **f** Trajectory analysis for MPSCs to IHCs and IPhCs, and for LPSCs to OHCs, DCs, IPCs, and OPCs. Upper panel color codes indicate cell types. Lower panel color codes indicate the time point of collection. MPSCs show a single bifurcation leading to IHCs or IPhCs. In contrast, while LPSCs transition to either DCs or OHCs, a clear bifurcation does not occur. HC hair cells, IPhC inner phalangeal cells/border cells, IPC inner pillar cells, OPC outer pillar cells, DC1/2 Deiters' cells rows 1 and 2, DC3, Deiters' cells row 3, HeC Hensen's cells, CC/OSC Claudius cells/outer sulcus cells, IdC interdental cells, ISC inner sulcus cells, KO Kölliker's organ cells, L.KO lateral Kölliker's organ cells, M.KO medial Kölliker's organ cells, OC90 OC90+ cells, L.PsC lateral prosensory cells, M.PsC medial prosensory cells. Scale bar in **e**, upper panel, 50 μm. Scale bar in **e**, lower panel, same for all cross sections, 10 μm.

expression of both *Tgfβr1* and *Fzd9* was observed in LPSCs at E16 and in non-HC derivatives of LPSCs; DCs and PCs, at P1. Consistent with the feature plots shown in Fig. 5b and Supplementary Fig. 10b, sporadic puncta were observed in other regions of the cochlear duct, suggesting limited expression of *Tgfβr1* or *Fzd9* outside the prosensory domain.

To determine whether *Tgfbr1* plays a role in development of LPSCs and/or their derivatives, cochlear explants were established at E14.5 and maintained in culture media containing either 20 μM of the Tgfβr1 antagonist SB505124 (see Methods section) or DMSO vehicle control for 5 days (DIV). Cultures were then fixed and labeled with antibodies against the HC markers MYO7A and POU4F3, and the progenitor/SC marker PROX1. We observed a significant loss of OHCs, but no change in PROX1+ SCs, in response to inhibition of Tgfβr1 (Fig. 5d). To determine whether inhibition of Tgfβr1 leads to cell death or an inhibition of OHC maturation, explants were treated with 20 μM SB505124 for 2 DIV, followed by an additional 3 DIV in control media. Results indicated a recovery of OHC formation (Fig. 5e), consistent with an inhibition of OHC maturation.

**Metabolic pathways may mediate Tgfβr1 effects in OHC.** The results presented above suggest a significant role for the Tgfβ pathway in development of OHCs. The specific downstream pathways that might be activated by Tgfβ in LPSCs are unknown. Tgfβ signaling has been shown to influence the activity of multiple pathways[47–49] to regulate different aspects of cellular development, including cell fate and differentiation. An extensive examination of all the known Tgfβ pathways in developing OHCs is beyond the scope of this study; however, we chose to examine a potential link between Tgfβ signaling and metabolism based on previous studies linking Tgfβ with metabolic pathways, such as the tricarboxylic acid (TCA) cycle[50], and the known links between hearing loss and metabolic/mitochondrial disorders[51].

To examine changes in the activity of specific pathways, expression of metabolic pathway gene sets[52] were compared between IHCs, OHCs, and pooled SCs at E16 and P1 (see Methods section, Supplementary Data 12) to generate an overall up (red) or down (blue) regulation map (Supplementary Fig. 11). Previous studies have demonstrated changes in glutamine metabolism and TCA cycle activation in response to Tgfβ signaling[50] and upregulation of both of those pathways was observed in OHCs. To examine which genes within those pathways might be upregulated in OHCs, violin plots were generated for each gene within the glutamine metabolism and TCA cycle GO annotations (Supplementary Fig. 11). Within the glutamine metabolism gene list, *Asparagine synthetase (Asns)*, *Glutaminase (Gls)*, and *Phosphoribosyl pyrophosphate amidotransferase (Ppat)* were specific to HCs (Supplementary Fig. 11),

while from the TCA cycle gene list, *Fumarate hydratase* (*Fh1*), *Dihydrolipoamide S-succinyltransferase* (*Dlst*), *Dihydrolipoamide S-acetyltransferase* (*Dlat*) were specific for HCs, and *Oxoglutarate dehydrogenase* (*Ogdh*) was specific to OHCs (Supplementary Fig. 11). *Gls* is a known target of Tgfβ (ref. [50]) while deficiencies in *Asns* have been linked to hearing loss[53]. Similarly, while TCA cycle disruption has not been linked with OHC development, it has been implicated in age-related hearing loss[54]. Further exploration into each of the examined metabolic pathways and the genes within it will provide greater insight into the mechanisms involved in OHC development.

**Differentiation of IHCs and OHCs at P7.** To examine the postnatal development of HCs and other cells within the cochlear duct, 3011 cells were collected at P7 (Supplementary Data 1). The cochlear sensory epithelium is not fully mature at this time point, but we identified this age as the best compromise between maturity and our ability to successfully dissociate and capture a significant number of cells. Unbiased clustering identified 12 clusters of cells (Fig. 6a) and examination of gene expression within those clusters allowed identification of all the known cochlear cell types (Supplementary Data 2). Consistent with the gradual degeneration of KO, the total number cells that were mapped to this region was decreased relative to younger ages and only three clusters were identified. Differential expression analysis identified known and candidate genes expressed in each cell type (Fig. 6b, Supplementary Data 1). IHCs and OHCs, which had been clustered as a single group in the whole cochlea P1 data set, appear as separate groups at P7. In contrast, DCs, which were in two clusters at P1, formed only a single cluster at P7.

To compare transcriptional maturity of IHCs and OHCs at P7 with functional IHCs and OHCs, we compared differential transcriptional expression between the 83 IHCs and 180 OHCs isolated at P7 with previously published HCs isolated from mature cochleae. Figure 6c illustrates the top ten DE genes between P7 IHCs and OHCs. To determine how many of those genes show similar patterns of expression in more mature tissues, we used the gEAR (https://umgear.org/) compare tool to generate scatter plots for gene expression in IHCs vs OHCs from three different cell-type-specific data sets collected at later time points (P15 or >P28)[25,55,56]. The top ten DE genes for IHCs and OHCs from P7 were mapped onto each of those scatter plots (Fig. 6d). Using a twofold difference in gene expression as a threshold, there was reasonable concurrence between the data sets. For IHCs, eight of the ten DE genes also showed greater than twofold expression in IHCs from the two studies that used RNAseq. Similarly, for OHCs seven of the ten genes that were DE at P7 also met the twofold threshold in the RNAseq data sets. Concurrence was not as consistent for Liu et al.[25], which was

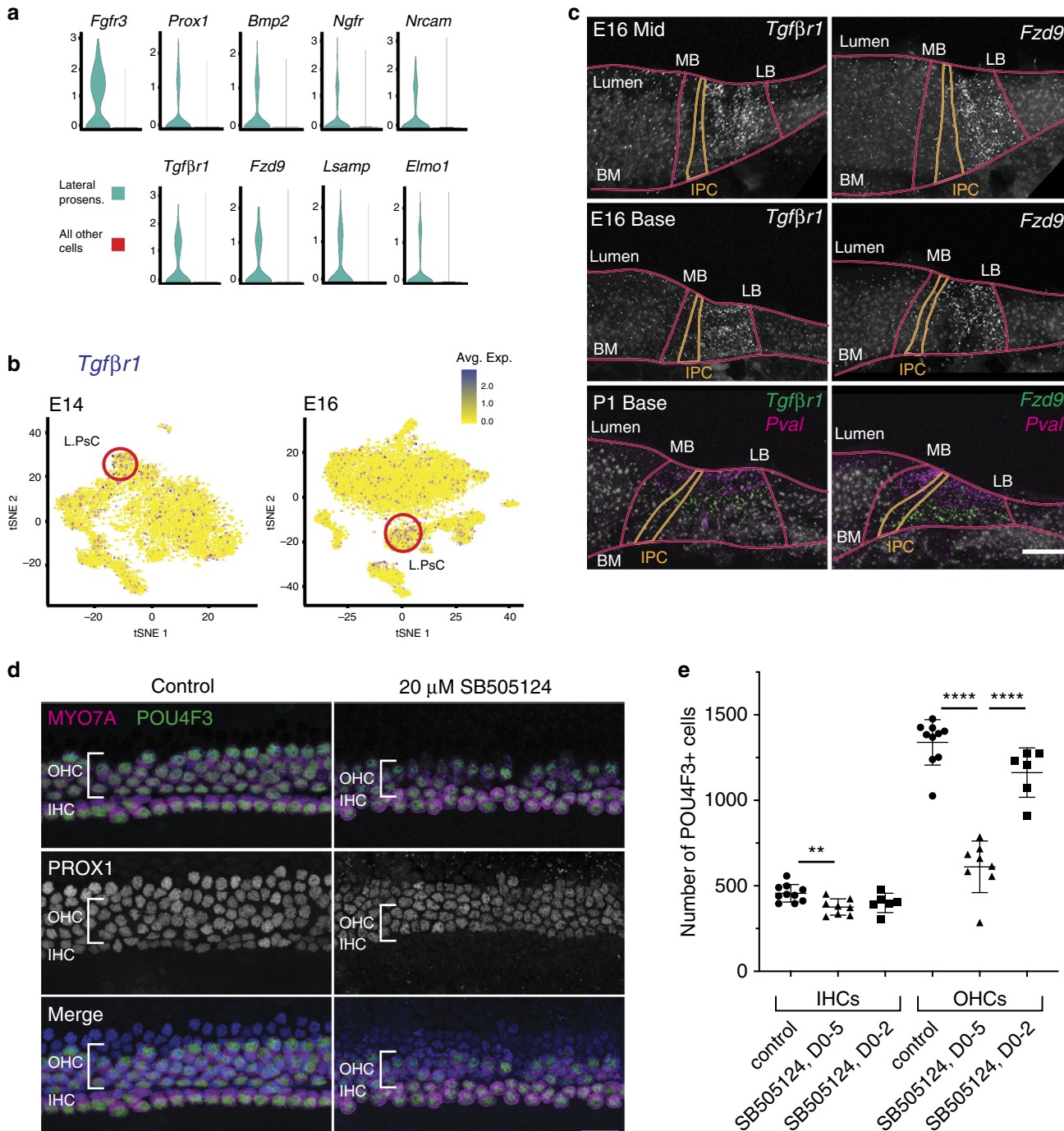

**Fig. 5 Tgfβr1 activity is required for formation of the lateral prosensory domain. a** Violin plots for five genes, *Fgfr3*, *Prox1*, *Bmp2*, *Ngf*, and *Nrcam*, that had previously been reported to be expressed in the lateral prosensory domain. Bottom row illustrates examples with similar patterns of expression that have not been previously reported in prosensory cells. **b** Feature plots for *Tgfβr1* at E14 and E16. Expression is concentrated in the LPsCs (red circles). See Fig. 4a, b for cluster identities. **c** smFISH confirms expression of *Tgfβr1* and *Fzd9* only in the lateral prosensory domain. Red lines indicate lumenal surface (lumen), basement membrane (BM), and approximate locations for medial (MB) and lateral (LB) boundaries of the OC. Approximate positions of IPCs (gold). At P1, expression of both *Tgfβr1* and *Fzd9* (green) is restricted to lateral SCs. HCs are marked with *Pvalb* (magenta). Minimum of two samples per probe with similar results. **d** Cochlear explants established on E14 and treated with either DMSO (control) or the Tgfβr1 antagonist SB505124 at 20 μM for 5 DIV. HC cytoplasm (anti-MYO7A, magenta) and nuclei (anti-POU4F3, green) and supporting cell nuclei (anti-PROX1, white or blue in merged image in bottom row) are labeled. Development of the lateral region of the OC is clearly disrupted (bracket), while the medial (IHC) region appears relatively unaffected. **e** Quantification of HC numbers per explant indicates a significant decrease in OHCs following 20 μM SB505124 treatment for 5 DIV (DO-5). A slight, but significant decrease in IHCs was also observed. Explants treated with 20 μM SB505124 for only 2 DIV (DO-2) followed by three additional days in control media did not show significant decreases in IHCs or OHCs. Data are presented as mean values ± SD, **$p = 0.0093$, ****$p < 0.0001$. $n = 24$ biologically independent explants (ten control, eight SB505124, DO-5, six SB505124, DO-2) with a total of $n = 35,217$ HCs counted over three separate experiments. Statistical test is one-way ANOVA followed by Tukey's multiple comparison's test. Source data are supplied as a Source data file. Scale bars in **c** and **d**, 20 μm.

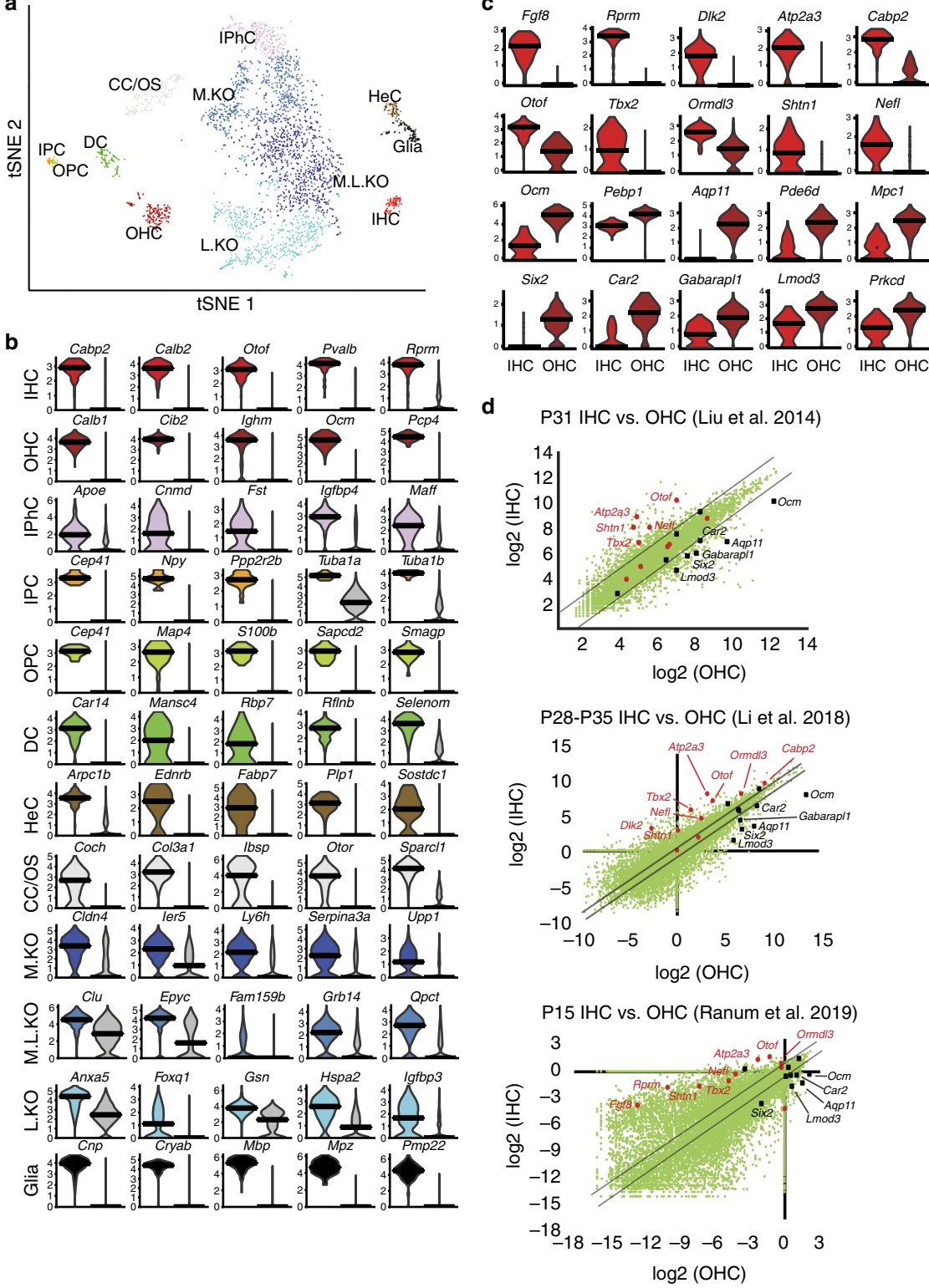

**Fig. 6 Comparison of gene expression between IHCs and OHCs at P7 and in adults. a** tSNE plot for ~3400 cochlear cells isolated from three separate experiments at P7. Specific clusters for different cells types are labeled. **b** Violin plots showing relative expression of the top five differentially expressed (DE) genes for each cell type (color coded as in **a**) by comparison with all other P7 cells (gray on the right in each graph). **c** Violin plots for the top ten DE genes between P7 IHCs and OHCs. **d** Scatter plots comparing gene expression in IHCs vs. OHCs collected at the indicated ages in the indicated publications. Gray lines mark cutoffs for twofold differences in gene expression. Red and black dots indicate DE IHC and OHC genes, respectively, from P7. Names are included for those genes located outside of the twofold threshold. IHC inner HCs, OHC outer HCs, IPhC inner phalangeal cells/border cells, IPC inner pillar cells, OPC outer pillar cells, DC Deiters' cells, HeC Hensen's cells, CC/OS Claudius cells/outer sulcus cells, M.KO medial Kölliker's organ cells, M.L.KO medial–lateral Kölliker's organ cells, L.KO lateral Kölliker's organ cells.

based on microarray rather than RNAseq. To further compare differential gene expression between the data sets, we compared the expression of all genes in the P7 data set that were DE in IHCs or OHCs and had an adjusted *p*-value of <0.01 (Supplementary Data 13). The resulting 146 genes for IHCs and 163 genes for OHCs were then mapped onto the scatter plot for the Li et al. data[55] (Fig. 6). For clarity, genes that fell below the twofold cutoff were excluded. Of the resulting 64 genes for IHCs and 77 genes for OHCs, 87.5% of IHC genes and 90.9% of OHC genes mapped on the predicted side of the scatter plot. However, if genes that mapped below the twofold threshold are also included, the percentages drop to 38.8% for IHCs and 42.9% for OHCs. Overall these results demonstrate both the challenges and limitations of comparing data sets from different studies. In addition, the findings suggest that while IHCs and OHCs are not fully mature by P7, their transcriptional profiles do share many similarities with functional IHCs and OHCs collected at later time points.

**Localization of deafness genes in individual cell types.** Average levels of expression of known and potential deafness genes were visualized within each cell type in the E14, E16, P1, and P7 data sets (Fig. 7). This analysis served to both confirm the quality of the data set and also to localize candidate deafness genes to specific cell types. Consistent with previous reports *Myo6*, *Myo7a*, *Myo3a*, *Cdh23*, *Pcdh15*, *Gipc3*, and *Cib2* (ref. [57]) were primarily expressed in HCs with many showing increased intensity of expression as HCs mature (Fig. 7; arrows). Similarly, *Gjb2* and *Gjb6* were most highly expressed in inner phalangeal cells and cells within KO (ref. [58]). Finally, we localized expression of genes that have been linked to age, noise or cisplatin-related hearing loss by GWAS but have not been studied extensively in the inner ear (https://www.ebi.ac.uk/gwas/). Many of these genes were most strongly expressed in HCs, but a few, such as *Ccbe1* and *St6gal-nac5* showed specific expression in IPCs or Claudius cells, respectively (Fig. 7; arrowheads).

## Discussion

We present an extensive data set, including over 30,000 cochlear single cells collected at four developmental time points. Identification of known cell types and validation by examination of known gene expression confirms the overall quality and comprehensiveness of the data. Moreover, because the OC develops heterochronically[59], this data set most likely contains cells that span the developmental spectrum from early progenitors through nearly mature cell types. While our analysis was necessarily limited to just a few cell types at a few time points, similar examinations of many other cell types and developmental transitions are clearly possible. The entire data set is available through the gEAR Portal (https://umgear.org/p?l=f7baf4ea).

Inner and outer HCs were known to be transcriptionally distinct by E16 (ref. [23]). In contrast, the timing and degree of transcriptomic differences between SC types was less obvious. The results presented here demonstrate that transcriptionally unique classes of SCs can be identified as early as E16, at which point both IPCs and IPhCs exist as discrete clusters of cells. Consistent with our previous work[4], IPhCs appear to share greater transcriptional similarity with cells located in KO than with other SCs. At P1, OPCs shared a high degree of transcriptional similarity with first/second row DCs. This result is consistent with previous work demonstrating plasticity within the LPsC population through at least the first postnatal week[60]. HeC and CC cells appeared as discrete cell clusters as early as E16, and fate mapping of *Fgfr3*+-lateral progenitors indicated that cells from this population develop as HeC/CC cells <0.15% of the time. These results suggest that HeCs and CCs should not be

considered as SCs, or at least not as SCs arising from the same precursor population as DCs and pillar cells.

Transcriptomes for OHCs isolated at P1, E16 and, to a lesser extent, at E14, indicated the presence of OHCs at different stages of developmental maturation. In contrast, HCs isolated at P7 appeared to be transcriptionally more uniform suggesting that a significant component of OHC development may be completed by P7. Using Monocle, we were able to create pseudotime trajectories for OHC development that suggested four possible phases of gene expression. Consistent with their known roles in overall HC development, genes expressed in early phases, such as *Sox2*, *Atoh1*, *Lhx3*, and *Pou4f3*, were expressed in both OHCs and IHCs. However, at later stages, genes that were unique to the OHC lineage, such as *Insm1*, *Neurod6*, and *Ikzf2*, were identified[23,61]. These genes are particularly intriguing as their timing of expression suggests a possible role in specification of OHC fate or repression of an IHC fate[23,61]. Genes expressed in subsequent phases of OHC development may regulate or act directly in different aspects of HC function, such as calcium regulation (*Atp2b2*) or buffering (*Calb2*), mechanotransduction (*Cib2*, *Tmc1*), or synaptic transmission (*Otof*)[5,6,29]; however, the roles of the genes that comprise these phases have not been fully examined.

OHCs isolated at P7 appeared to be transcriptionally homogenous. This result is consistent with the ongoing maturation of HCs during the first postnatal week but also suggests that the overall depth and complexity of this analysis was not sufficient to resolve the graded changes in HC phenotype and function that are known to occur along the tonotopic axis[62]. Similarly, DCs and pillar cells, which also show graded changes along the tonotopic axis, were transcriptionally similar at P7. The overall number of both HCs and support cells in the P7 data set was relatively low, so it is possible that increasing the number of cells analyzed at P7 and increasing the sequencing depth might allow for better resolution of cellular differences.

All of the cells that comprise the OC are thought to arise from a common prosensory progenitor population[63]. While definitive lineage-tracing data specifically examining the fates of individual prosensory cells is limited, clonal analyses suggest that prior to E14 prosensory cells have the potential to generate daughter cells that can develop as any cell type within the OC (ref. [64]). However, unbiased clustering of single cells isolated at E14 indicated the presence of two clusters of cells that were positive for the prosensory markers *Sox2* and *Cdkn1b* (refs. [65,66]). Several genes, including *Fgfr3* and *Prox1*, which are known to be restricted to the lateral domain[39,40], were found to be expressed exclusively in one of these clusters, suggesting that it represents the lateral prosensory domain. Fate mapping of this group using *Fgfr3*[icre] indicated a strong bias toward cell fates located within the lateral domain of the OC. While fate mapping during normal development is not synonymous with fate restriction, these results do suggest that *Fgfr3*+-prosensory cells are strongly biased toward lateral fates. This conclusion is also supported by the observation that defects in *Fgfr3* mutant mice are limited to OHCs, PCs, and DCs. Also consistent with an early medial–lateral division of the prosensory domain is the observation that strongly *Atoh1*+ HC precursors appear to develop in two spatially distinct columns of cells, located in the medial and lateral prosensory regions[67]. Given that these regions are transcriptionally distinct prior to HC formation, these results would suggest that while IHCs and OHCs share a high degree of transcriptional similarity, their lineages are separate.

Analysis of the scRNAseq data for LPsCs identified a number of genes that are restricted, within the cochlea, to this group. *Tgfβr1* was particularly intriguing, as hearing loss has been reported in several syndromes that arise from mutations in

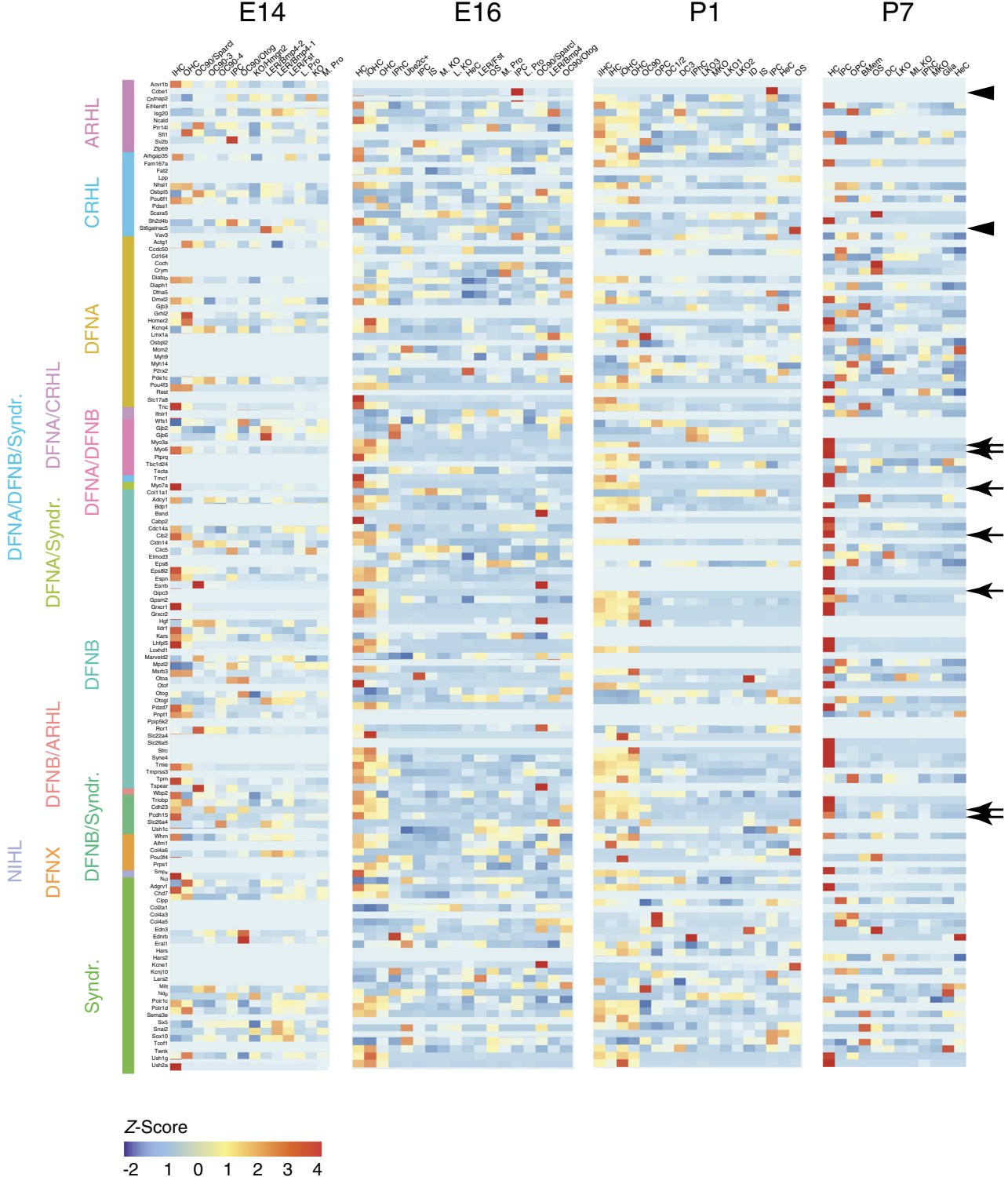

**Fig. 7 Localization of deafness genes.** Heat maps illustrating cell-type-specific expression (as a *z*-score for cell-type-averaged expression) for deafness-related genes in cochlear cell clusters from E14, E16, P1, and P7. Cluster identities for each time point are indicated along the top of each heat map and disease classes are indicated on the *Y*-axis. Consistent with previous work, many deafness genes that are known to be expressed in HCs, such as *Myo6*, *Myo7a*, *Myo3a*, *Cdh23*, *Pcdh15*, *Gipc3*, and *Cip2*, show exclusive expression in HC clusters in particular at P7. Other genes that have been associated with hearing loss but not examined within the inner ear show expression in other cell types. For example, *Ccbe1*, which when mutated can lead to Hennekam syndrome, is only expressed in IPCs. ARHL age-related hearing loss, CRHL cisplatin-related hearing loss, NIHL noise-induced hearing loss, Syndr. syndromic.

*TGFβR1* (refs. [2,3]). Inhibition of Tgfβr1 disrupted development of OHCs, but did not influence the expression of PROX1. This result suggests that Tgfβr1 plays a role in the maturation of OHCs, which are derived from the lateral prosensory domain, rather than in the specification of these cells. Given the importance of the development of cell types that are derived from the lateral prosensory domain, an understanding of the factors that specify this group of progenitors could have significant implications for understanding cochlear function and the design of regenerative strategies. Since several markers of the lateral prosensory domain were already present at E14, it seems possible that specification of this region could occur at an even earlier time point. Consistent with this hypothesis are several studies that have examined the interaction between *Fgf20* and *Fgfr1* (refs. [38,68]). Both genes are expressed in the developing cochlear duct at least as early as E11.5 (refs. [38,69]) and deletion of either gene leads to a loss of lateral domain cells, including OHCs, DCs, and PCs (refs. [38,68]). By E13.5, *Fgf20* expression appears to be restricted to MPsCs while *Fgfr1* expression is restricted to LPsCs, suggesting that a paracrine interaction between the two cell groups acts to induce key aspects of the lateral domain of the OC.

In conclusion, this study presents a single-cell RNAseq atlas for the developing cochlear epithelium at different embryonic and early postnatal time points. Validation of the results suggests that these data will provide a valuable resource for the examination of multiple developmental events during the formation of the mammalian cochlear duct and OC.

## Methods

**Isolation of cochlear cells.** Timed-pregnant CD1 females were obtained from Charles River and maintained within the Porter Neuroscience Research Center Shared Animal Facility. All animal care and housing was conducted in accordance with the NIH guidelines for animal use (Protocol 1254-18). For each embryonic time point, a pregnant female was euthanized, and cochleae were dissected from 10 to 12 pups of either sex. For postnatal time points, animals were euthanized and cochleae were dissected from ~5–8 animals of either sex. Cochlear ducts were dissected and placed in DMEM/F-12 with 0.2 mg/ml of thermolysin and 10 kunitz/ml of DNase I for 10 min at 37 °C. Following the incubation period, stromal cells and the cochlear roof and lateral wall were dissected away to isolate the epithelial floor of the cochlear duct. Cochlear floor epithelia were combined in a single tube and incubated in 0.25% trypsin-EDTA for 15 min at 37 °C with gentle trituration every 5 min. At the end of the incubation, trypsin was inactivated by adding an equal volume of fetal bovine serum and dissociated cells were then passed through a 40 μm strainer, pelleted at $300 \times g$ and then resuspended in 10–15 μl of DMEM/F12 with 10% fetal bovine serum[4]. A minimum of three biological replicates, each of which represents an independent collection of single cells from a single time point as described above, were included for each time point.

Single cells were captured and lysed, and mRNAs were reverse transcribed into cDNAs using a 10X Genomics Chromium Controller. cDNA libraries were prepared using Chromium Single Cell 3′ Reagents following the manufacturer's instructions. Libraries were sequenced on an Illumina NextSeq to generate 60 bp of sequence to identify transcript identity. Sequences were aligned to the Ensembl mouse MM10 assembly using Cell Ranger 2.1.1 analysis software (10X Genomics).

**Data preprocessing, dimensionality reduction, clustering, and visualization.** Processing of the Cell Ranger output data was done with Seurat (R package v2.0; https://github.com/satijalab/seurat)[70]. First, Seurat's "Read10x" function imported the Cell Ranger output as cell-by-gene counts expression matrices. Genes in at least ten cells were included in the analysis. Cells with <200 unique genes and 1500 unique molecular identifiers (UMI) or >3000 unique genes and 15,000 UMI were excluded from the analysis. Cells with >5% mitochondrial genes or >5% stress genes present were excluded from downstream steps. After processing, 30,670 cells (out of 58,143) were included in the final analyzed data set (Supplementary Data 14). The expression data were then log transformed, normalized, and scaled for sequencing depth. UMI, mitochondrial content, and stress gene content scores were "regressed-out" using Seurat's "ScaleData" function. Seurat's canonical correlation analysis (CCA) accounted for batch effects between expression data sets from the same time point and merged these matrices to create a new object for each time point (http://github.com/kelleylab/cochlearSEscrnaseq).

Statistically significant canonical correlation components identified by Seurat's "RunCCA" function were used to define the dimensions for the *t*SNE nonlinear dimensionality reduction analysis, which then visualized the cells on a 2D *t*SNE plot. Unsupervised clustering identified groups of molecularly distinct cells on the plot. Clustering was done by testing a range (0.2–2.4 in 0.2 increments) of values

for the "resolution" parameter in Seurat's "FindClusters" function. The average OOB error was calculated of each run using Seurat's "AssessNodes" function. The resolution with a low OOB error and high cluster number was chosen.

Cells on the *t*SNE plot were annotated based on cluster specific genes identified by DEsingle (R package v1.4.0; (https://github.com/miaozhun/DEsingle))[71] and Seurat's "FindAllMarkers" (min.pct = 0.25, thresh.use = 0.25) differential expression analysis (http://github.com/kelleylab/cochlearSEscrnaseq). Cell-type-specific markers identified in Seurat are listed in Supplementary Data 2.

**Comparisons of cell-type-specific gene expression with previous publications.** The gEAR Portal (https://umgear.org/) "Dataset Comparison Tool" was used to generate scatter plots of gene expression between specific cell types from previously published data sets. For the P1 comparison, cell clusters classified as HCs and support cells from Burns et al.[4] were plotted. For the P7 comparison of inner and outer HCs, inner and outer HCs from the Liu et al.[25], Li et al.[55], and Ranum et al.[56] data sets were plotted. DE genes between comparable cell types in this study were determined using Seurat. For the P1 comparison, inner pillar cells and all Deiters' cells were combined to form a single support cell group. This was the most comparable comparative group based on the level of GFP expression in those cells in Burns et al.[4]. The locations of the top 50 (for P1 HCs and SCs) or top 10 (P7 IHCs and OHCs) were then mapped onto those scatter plots.

**Inferring cellular localization of hearing loss genes.** Hereditary hearing loss genes were obtained from https://hereditaryhearingloss.org/. Genes associated with acquired hearing loss were taken from the NHGRI-EBI GWAS Catalog (https://www.ebi.ac.uk/gwas/). Cluster averages of *z*-score adjusted normalized gene expression were taken. Heat maps visualize the *z*-scores of the cluster averages.

**Monocle trajectory analysis.** HC trajectory analyses were done with Monocle (R package v2.0; (https://github.com/cole-trapnell-lab/monocle-release))[26] and DEsingle. Preprocessed Seurat objects were imported into Monocle with the "importCDS" function. Monocle's "orderCells" function arranged cells along a pseudotime axis to indicate their position in a developmental continuum. Then, Monocle's differential expression (DE) analysis identified genes that significantly varied in expression along the pseudotime axis applying a false discovery rate of 0.1. The parameter provided for the DE analysis was either "cell type" for the P1 outer HC trajectory or "time point" for the E14–P1 trajectories. For the outer HC trajectory, the resulting genes were clustered into four groups to represent four distinct gene expression trends along the continuum (http://github.com/kelleylab/cochlearSEscrnaseq). Monocle was used to visualized gene expression trends along the trajectories.

**Identifying cell-type-specific regulons.** A modified SCENIC (https://github.com/aertslab/SCENIC; v0.99)[30] pipeline described in ref. [72] was used to infer TFs driving the maintenance of CE cell states. First, randomly selected subsets of 5–20 cells within each cell type were pooled. SCENIC was run on the average expression of these modified data sets to find significantly enriched gene regulons and regulon activity scores (RAS). Then, RAS scores for each regulon were converted to regulon specificity scores (RSS) using the Jenson–Shannon entropy-based divergence metric[72]. Regulons were then ranked by their RSS to identify highly specific gene regulons for each CE cell type (http://github.com/kelleylab/cochlearSEscrnaseq).

**Quantifying cell-type-specific metabolic dynamics in CE development.** Time point and cell-type-specific metabolic markers[73] were identified using Seurat's FindAllMarkers function (adj. $p < 0.05$; Wilcoxon rank-sum test). VennDiagram ((https://github.com/cran/VennDiagram); v1.6.20) was used to visualize pooled cell-type-specific metabolic markers.

Cell-type-specific metabolic pathway activity dynamics were inferred from intra-pathway variation in gene expression over time. Pathway gene sets were provided by ref. [52].

First, Seurat's FindMarkers function was used to find genes that are DE between P1 and E16 time point cells for each metabolic pathway *n*. The frequency $p(U_n)$ and $p(D_n)$ of up-and-down regulated genes was determined by:

$$p(U_n) = \frac{U_n}{U_n + D_n}$$

$$p(D_n) = \frac{D_n}{U_n + D_n}$$

where $U_n$ represents the number of genes upregulated in expression in P1 cells (avg. logFC > 0; $p < 0.05$) and $D_n$ represents the number of genes downregulated in expression in P1 cells (avg. logFC < 0; $p < 0.05$).

Then, vectors $P^U = (p_1^U, p_2^U, ...., p_n^U)$ and $P^D = (p_1^D, .., ...., p_n^D)$ were created to represent the distribution of DE gene frequencies. Both vectors were normalized so that $\sum_{i=1}^{n} p_i^U = 1$ and $\sum_{i=1}^{n} p_i^D = 1$.

The regulation matrix was calculated by subtracting $P^D$ from $P^U$. Negative values were colored blue, and positive values were colored red. The matrix illustrates the predicted switches in metabolic pathway dependence from E16 to P1.

**Localization of gene expression by smFISH.** Gene-specific probes and the RNAscope® Fluorescent Multiplex Reagent Kit (320850) were ordered form Advanced Cell Diagnostics. Cochleae were collected from CD1 mice of both sexes at embryonic day 16 (E16) or postnatal day 1 (P1), fixed in 4% paraformaldehyde overnight, and then cryoprotected through a sucrose gradient (5%, 10%, 15%, 20%, and 30%). Samples were then embedded in Tissue-Tek O.C.T compound, and sectioned on a cryostat at 10 μm thickness. Hybridization protocol was carried out based on the manufacturer's suggestions. All fluorescent images were obtained on a Zeiss LSM 710 confocal microscope.

**Fate mapping of Fgfr3+ cochlear cells.** Fgfr3$^{icre/ERT2}$, Gt(ROSA)26Sot$^{tm14(CAG-tdTomato)Hze}$ (R26R$^{tdTom}$), and xGt(ROSA)26Sor$^{tm4(ACTB-tdTomato,-EGFP)Luo}$ (R26R$^{mT/mG}$) mice were ordered from Jackson Laboratories. Animals were crossed to generate Fgfr3$^{icre}$/R26R$^{tdTom}$ or Fgfr3$^{icre}$/R26R$^{mT/mG}$ double heterozygous mice. Tamoxifen was resuspended in flax seed oil to a stock concentration of 10 mg/mL. On the day of gavage, a working stock of 2.5 mg/mL tamoxifen plus 20 mg/mL progesterone was dissolved in flax seed oil. A total of 100 μL of tamoxifen solution was administered via oral gavage to each pregnant dam for a final dose of 250 μg tamoxifen per animal. All pregnant dams were given a total of one injection on E14, E15, or E16. Animals were maintained until E18 or P1 and then cochleae from animals of either sex were dissected, fixed in 4% paraformaldehyde in 1× PBS, and stained with anti-tdTOMATO and phalloidin to label filamentous actin. Recombined cells were identified based on expression of td-Tomato and cell types were determined based on morphology.

**Inhibition of Tgfβr1 in vitro.** Timed-pregnant CD1 females were euthanized at E14, embryos of either sex were removed, and cochleae were dissected and established as explant cultures. Briefly, cochlear ducts were isolated from the surrounding otic capsule and then separated from the vestibular structures. The roof of the duct was removed to expose the developing sensory epithelium and each explant was then adhered to a Matrigel-coated coverglass[74]. Explants were maintained in DMEM with 10% fetal bovine serum at 37 °C in 5% CO$_2$. To inhibit Tfgβr1 activation, experimental explants were treated with 20 mM SB505124 (SigmaAldrich) in 0.1% DMSO[75,76] for 2 or 5 days beginning on day of culture. Controls were treated with just 0.1% DMSO. Explants were then fixed and immunolabeled to examine HC and SC development using anti-MYO7A (HC cytoplasm), anti-POU4F3 (HC nuclei), and anti-PROX1 (lateral SC nuclei). The total number of POU4F3+ HC nuclei in the explants were counted, and statistical comparisons were made using ordinary one-way ANOVA followed by Tukey's multiple comparisons test in Prism GraphPad.

**Reporting summary.** Further information on research design is available in the Nature Research Reporting Summary linked to this article.

## Data availability

The authors declare that all data supporting the findings of this study are available within the article and its Supplementary Information files or from the corresponding author upon reasonable request. Single-cell gene expression data have been deposited in the Gene Expression Omnibus data repository under accession code: GSE137299. Gene by cell expression matrix and data visualizations presented in this paper are available through the gEAR Portal (https://umgear.org/p?l=f7baf4ea). The source data file includes data relevant to data presented in Fig. 4e (Fgfr3 fate mapping) and Fig. 5c (effects of inhibition of Tgrbr1 on outer HC development).

## Code

Analysis was done in R (version ≥ 3.4.1). Code is available on GitHub (http://github.com/kelleylab/cochlearSEscrnaseq).

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

## Acknowledgements

The authors would like to thank the outstanding assistance of the Genomics and Computational Biology Core at the NIDCD. The authors also wish to thank the staff of the Shared Animal Facility, Porter Neuroscience Research Center for outstanding animal care. The authors also wish to thank Drs. Lisa Cunningham, Doris Wu, Lisa Goodrich, and Brikha Shrestha for providing comments on an earlier version of this manuscript. This research was supported by funds from the NIDCD Division of Intramural Research to M.W.K. (DC000059) and the GCBC (DC000086), from King's College CCRB to ZFM and by NIDCD/NIH (R01DC013817), NIMH/NIH (R24MH114815) and the Hearing Restoration Program of the Hearing Health Foundation to R.H. This work utilized the computational resources of the NIH HPC Biowulf cluster (http://hpc.nih.gov).

## Author contributions

L.K. analyzed data and results, generated figures, and wrote and edited the manuscript; M.C.K. generated and analyzed data, and edited the manuscript; Z.F.M. generated and analyzed data, and generated figures; A.A.-R. generated data and figures; K.E. and A.L. generated and analyzed data; A.T.P., K.S.S., and J.C.M. generated data, J.O. analyzed data, J.C.B. generated and analyzed data, and edited the manuscript; R.H. analyzed data and edited the manuscript; E.C.D. generated data and edited the manuscript; and M.W.K., analyzed data, generated figures, wrote and edited the manuscript.

## Competing interests

The authors declare no competing interests
