## [Peer Review File · Nature Communications]

Reviewers' Comments:

Reviewer #1:

Remarks to the Author:

The paper provides a novel set of data from E14.5 to postnatal P31 of a novel RNAseq atlas. The provided assessment will be of great interest in the community and provides a concise overview. Further strengthening the current overview could be exploring somewhat the differential growth of the Organ of Corti to expand current research to influence the field.

Overall, this assessment is well-done, with great progress in the right direction toward cochlear hair cells. This paper provides novel insights that compiled in an excellent single cell RNAseq level analysis of E14.6 and older stages of 30,000 cells. The main insights are careful and suggested that OHC cells arise from a progenitor pool.

P132 and 144 Previous work showed a progression of Atoh1 that expanded until at P1 (Matei et al., 2005) and seem to propose some differentiation (Dabdoub et al., 2009). This work showed the basal-to-apical axis (Mateo et al., 2005) that contrasted to the apex-to-base Prox1 expansion (Fritzscht et al., 2010). How precise could the different increase of Atoh1 progression is made in the apical regions of P1, mentioned as 'different stages of maturity'? Data supplies different proliferation of all hair cells and highlight the effects of proliferation driven by cell cycle progression by n-Myc (Kopecky et al., 2011, Dominguez-Frutos et al., 2011) also claimed c-Myc involvement (Kwan et al., 21015). Beyond providing a catalog of genes that are expressed OHCs, it would be worthwhile to correlate the earlier stages with forming the future directions driven by proliferation of precursors, clearly beyond the current assessment.

P171 showed highly delayed hair cell degeneration that also shows a correlative loss of sensory innervation in parallel to the base-to-apex progression (Xiang et al., 2003). The surprisingly long retention of hair cells showed at least 6 months. This data suggesting a long-term survival of Pou4f3 null mice beyond some hair cell, also with a delay the innervation retains.

E196 Recent data shows that Sox2 early deletion causes entirely blocks hair cell formation but also found a differential effect of vestibular versus cochlear hair cells (Dvorakova et al., 2019). Given that all cochlear differentiation is absent in these mice it points into a much earlier loss of initial HC induction.

P366 Beyond the general base-to-apex of Atoh1 (Mateo et al., 2005) and apex-to-base Prox1 (Fritzscht et al., 2010) is following a counter-differentiation of BDNF (apex-to-base) and Nt-3 (base-to-apex; Farina et al., 2001). Clearly certain projects follow the apex-to-base progression (Chena and Segil, 2002; Kopecky et al., 2011). How is that differentiation progressing from Atoh1 in a counter-direction and why is it in similar NT-3 expression?

Reviewer #2:

Remarks to the Author:

Overall comments

- The authors present a cohesive, in-depth analysis of developing mouse cochlear cell transcriptomes. The results confirmed distinctive expression profiles of cells derived from lateral and medial progenitor cells. Additionally, developmental trajectories of hair cells identified genes that may play an important role in cell fate specification and development. Collectively these data provide a valuable resource for characterizing the transcriptional changes that occur during development of the highly distinctive cochlear inner and outer hair cells. This provides a roadmap for eventual identification of genes and regulatory pathways that control inner and outer hair cell specification and development. The manuscript is nicely written.

- The total number of cells included in the study (for all time points?) was 30,000. It is unclear how many cells were collected and analyzed at each time point, and what proportion of these cells

were representative of the hair cell (immature or developing) populations. The lack of clarification of the number of cells analyzed at each time point may be construed as misrepresentation of data and should be addressed.

- Burns et al. (2015, Nature Communications 6: 8557) published single-cell RNA-seq transcriptomes of cells from neonatal inner ear. Comparison should be made between the two datasets to determine how consistent are the gene expression profiles of different cell types, especially the hair cells.

Specific comments

- There are a considerable number of abbreviations used throughout the manuscript, please include a comprehensive list of abbreviations at the end.

- Remove "already" from line 78.

- Line 130, 159, 180- "see below" is a vague reference. I suggest removing these to avoid confusion.

- After supplemental figures 5 & 6 are referenced in the text, the numbering in the text does not correspond with the numbered figures (missing 7). Please confirm that the numbering of all supplemental figures is correct, both in the text and the attachment.

- Line 241- change to "Fig. 5c".

- Line 244- change to "Fig. 5b".

- Line 275- add "expression of metabolic pathway gene sets".

- Line 311- change "single cell" to "cell type-specific" to avoid implying that all the other data sets are single cell RNA-seq, as two of them are not.

- While the justification for the discrepancies between datasets is valid (line 322-327), it would be worth noting the number of cells identified as IHCs or OHCs in the dataset (out of the 30,000 cells analyzed) as it may be considerably less than the total number of cells that were analyzed by Liu et al. (2014, J Neurosci 34: 11085-95) and Li et al. (2018, Scientific Data 5:180199), though there is a reference to this in the discussion.

- Line 412- change "lineal" to "linear".

- What version of Cell Ranger was used?

- How many total cells were initially sequenced (at each time point)? What was the total number of cells included in the final analysis after applying exclusion parameters (at each time point)? There are references to cells numbers collected (see Fig. 1b. 14,000 cells at P1), while Fig. 3 does not include the number of HCs included in the analysis. The number of cells collected at E14 and E16 is unclear, while Fig. 7a states that 3,400 P7 cochlear cells were isolated.

- Line 473- remove extra "(" at end of sentence.

- What p-value FDR was used for Monocle analysis?

- Line 544 refers to reference 33 as bulk RNA-seq data, but Liu et al. 2014 was microarray data. It should be Li et al (2018) or Liu et al (2018, Front Mol Neurosci 11:356).

- Line 563- What was the dose of tamoxifen used? Was it an IP injection?

- Additionally, the methods should be written to clarify "Pregnant dams were given a single injection of tamoxifen" unless the individual embryos were injected?

- Line 571- From what supplier was the SB505124 obtained?

- Figures should be examined, and font sizes should be increased when possible.

- Figure 1d. Y-axis label? Relative expression quantified as fold change? Is this consistent throughout the other figures? Please clarify.

- Figure 3d. The data points for Tmc1 are not clearly visible, consider adding circles or arrows, or changing the color scale.

- Figure 3e. What do the red and blue circles denote?

- Figure 4c and d. The blue/red/green annotations are confusing. Clarify if the color corresponds to a dot, circle or both by modifying the legend.

- Figure 5d and e. Why were two different cell markers used (MYO7A and POU4F3) for the qualitative and quantitative analysis? There should be consistency across both analyses.

- Figure 5d. Add a scale bar.

- Figure 5e. The abbreviations are inconsistent with the text. Change to DIV-2 and DIV-5.

Reviewer #3:

Remarks to the Author:

The manuscript from Kolla and Kelly: "Characterization of cochlear hair cell development at the single cell level" provides a very exhaustive characterization of developmental transcriptional profiles, at the single cell level, for the cochlear sensory epithelium.

This is a useful data set for the research community, and will serve as an important reference for future studies. The authors provide developmental trajectories, for different cell populations, including cells in the Kölliker organ, lateral and medial progenitors within the cochlear prosensory domain and inner and outer hair cells.

The large amount of data generated by this approaches and presented here makes however the story in some parts unstructured, and seem to lack focus and depth, in particular for the more mechanistic aspects.

-One of the major findings is the identification, already at E14, of two transcriptionally distinct populations of Sox2 cochlear progenitors, further named lateral and medial progenitors. Analysis at different time points as well as pseudotime analysis seems to confirm that lateral progenitors will give rise either to supporting cells (Deiters) or outer hair cells, while medial progenitors, seem to contribute to inner hair cells and inner border/phalangeal cells.

It is not clear however from the trajectory analysis (lines 211-220; figure 4f and supplementary 9 (actually 10), nor legends and methods) if the populations were clustered a priori, before Monocle was used to define trajectories. If I understand correctly, the analysis was performed separately on (IHC, MPSCs and IPhC) or (LPC, DC, OHC)?.

If this is the case, the authors should show what a more unsupervised approach looks like, where medial and lateral progenitors and all deriving cell types are analyzed for the 3 time points. Ideally a earlier time point (E13? Should be included). It may be worth to expand figure 4f by combining data from supplementary figure10 (referred to in the text as figure 9).

-Fgfr3 lineage tracing has already been done, giving similar results, using nevertheless later time points (from p1; Cox et al JARO2012) confirming the findings. Others have reported instead defects in outer hair cell formation in FGF20 KO animals (Huh Plos Biology 2012), or lack of all hair cells by blocking antibodies against FGF20 on explant culture. While FGF20 is expressed in medial progenitors, effect on OHC development should be discussed. These findings should be included in the discussion.

-The authors identify Tgf β 1 as a putative novel marker of lateral progenitors. The expression of Tgf β r does not seem to be exclusively restricted to lateral progenitors (as shown in the tSNE plots Fig 5b), but seems to have higher expression in this population. In addition, inhibition of Tgfb signaling using a small molecule inhibits outer hair cell formation in organotypic cultures. The following figure (Figure6) on metabolic changes and associated signature however does not show in any way the contribution of Tgfb signaling into the process, even though it is presented as one of the Tgfb effects. Despite being potentially interesting, a more detailed mechanistic analysis of the process (with Alk inhibitors or SMAD2 conditional deletions) would be needed to show how much these processes depend on Tgfb signaling. I would suggest removing entirely figure 6.

-The analysis of deafness-associated genes presented in Figure 2M breaks the story line. In my opinion it could be discussed in conjunction with the same analysis (performed at E14,16 and P7, and presented in supplementary figure 3) at the end of the manuscript.

As hair cell development is not the only feature presented, the title could be referring more generally to the development of the cochlear sensory epithelium.

Minor comments

Line 28-29 (Abstract)

... comprising several unique cell types... remove "including OHC, DC and PC", as all are unique cell types.

Line 111

Tbx expression by smFish :

Tbx2 is indicated as IHC specific and in the text it is mentioned: "it is detected at lower levels in supporting cells". Expression seems to be very high in the KO region, please rephrase.

OC90 positive population is indicated for different developmental stages but not discussed, it is not clear what it represents-> specify

OC90 is obviously very highly expressed at mRNA level in the cochlea as shown by several reports. Hartman et al (Frontiers in Neuroscience) reported in situ hybridization of OC90 and showed localization in the roof of the cochlear duct, however here tissue microdissection (even though not described in detail) seems to be excluding this compartment. What are these cells?

Supplementary tables

Excel Tables provided are not labeled as table 1,2,3 etc (as in the text)

This is the kind of file name visible to the reviewer:

224312_0_supp_4078389_pxsd4x_

it is very difficult to find the correct tables the authors refer to → rename

Supplementary figures

-Supplementary figure 7 is missing, or not existing and legends are therefore not correct from 7 onwards. In general, given the substantial amount of data presented in the supplementary material, figure legends should be extended, to better understand the content.

-What is the difference between data presented in supplementary Figure 10a and Figure 5a? From the legend it seems it's the same data. (violin plots for lateral prosensory domain)

-Supplementary figure 6 Tectorin b as highest gene in Lateral progenitors? Is this correct?

-Supplementary figure 9 (should be Supplementary figure 8)

The text in the manuscript does not reflect what is shown in the figure:

Fgf20 was expressed in MPCs, inner phalangeal cells and IHCs, while Fgfr3 and Prox1 were expressed in LPsCs, DCs, OPCs and OHCs (Suppl. Fig. 8).

There's no apparent expression of FgF20 in inner phalangeal cells in the violin plots;

There's no expression of Fgfr3 in OPC, the label indicates inner Pillars.

(line 195 in the text, same paragraph, Fgfr3 is labeled as Fgr3)

Figures

Figure 5 : Orange-shaded area over the smFish prevents seeing the expression of the probe in that area, leave the contour, remove the color.

Methods

The authors indicate that at embryonic time point they have isolated cochleas from 10-12 pups, while for postnatal tissue from 5-8 animals.

It is therefore not clear what the authors refer to with a "minimum of 3 biological replicates was used". I believe all cochleas from a single developmental stages pooled for analysis?
Please clarify

Kolla, Kelly et al. NCOMMS-19-31026

Please Note, all significant changes are indicated in **BOLD** text

Reviewers' comments:

Reviewer #1 (Remarks to the Author):

The paper provides a novel set of data from E14.5 to postnatal P31 of a novel RNAseq atlas. The provided assessment will be of great interest in the community and provides a concise overview. Further strengthening the current overview could be exploring somewhat the differential growth of the Organ of Corti to expand current research to influence the field.

Overall, this assessment is well-done, with great progress in the right direction toward cochlear hair cells. This paper provides novel insights that compiled in an excellent single cell RNAseq level analysis of E14.6 and older stages of 30,000 cells. The main insights are careful and suggested that OHC cells arise from a progenitor pool.

P132 and 144 Previous work showed a progression of *Atoh1* that expanded until at P1 (Matei et al., 2005) and seem to propose some differentiation (Dabdoub et al., 2009). This work showed the basal-to-apical axis (Mateo et al., 2005) that contrasted to the apex-to-base *Prox1* expansion (Fritzscht et al., 2010). How precise could the different increase of *Atoh1* progression is made in the apical regions of P1, mentioned as 'different stages of maturity'?

The point that the reviewer would like to make is a bit unclear here. We are confident that the linear arrangement of P1 OHCs reflects a developmental gradient for the following reasons. First, when we examined expression of genes that are known to be down regulated as hair cells develop, such as *Sox2* or *Cdkn1b*, those genes showed a down-regulation in the OHCs located on one side of the linear distribution of cells illustrated in Figure 3. In contrast, genes such as *Calb2* and *Tmc1*, which only turn on in more mature OHCs were restricted to the other side of the linear gradient. Second, work from our lab and others has demonstrated that *Atoh1* is expressed in a gradient, albeit a rapidly developing gradient, that begins in the base of the cochlea and extends towards the apex (Lanford et al., 2000; Chen et al., 2002; Fritzscht et al., 2011; Driver et al., 2013). The reviewer mentions some conflicting data regarding the onset of expression of *Prox1*. Fritzscht et al (2010) presented in situ hybridization data indicating a basal-to-apical gradient but also showed reporter activity that suggested an apical-to-basal onset. Bermingham-McDonogh et al (2006) show a clear basal-to-apical gradient of *Prox1* protein onset. However, regardless of the pattern of onset, the down-regulation of *Prox1* in hair cells clearly occurs in a basal-to-apical gradient (Bermingham-McDonogh et al. (2006). Therefore, as a further confirmation of the developmental gradient in the P1 OHC analysis, we generated feature plots for *Prox1* which shows a down-regulation along the same axis as *Atoh1*, as well as *Isl1* and *Fgfr3*, both of which have been shown to be down-regulated in OHCs as they mature. The feature plots below use a full spectrum scale to illustrate changes in gene expression. All four genes show gradients along the OHC linear distribution that are consistent with a

maturation gradient. Finally, a similar feature plot is shown for *Fzd9* which also shows a gradient along the same linear pattern. For *Fzd9* we also show smFISH results from the middle (less mature) and basal (more mature) regions of the cochlea. Consistent with the feature plot, there is a down-regulation of *Fzd9* in OHCs with maturation. Based on these data, we feel very confident about the developmental/maturation gradient that is present in the P1 OHC data set.

Data supplies different proliferation of all hair cells and highlight the effects of proliferation driven by cell cycle progression by n-Myc (Kopecky et al., 2011, Dominguez-Frutos et al., 2011) also claimed c-Myc involvement (Kwan et al., 2015). Beyond providing a catalog of genes that are expressed OHCs, it would be worthwhile to correlate the earlier stages with forming the future directions driven by proliferation of precursors, clearly beyond the current assessment.

We agree that it would be useful to correlate early stages of OHC development with proliferation but that it is beyond the scope of this work.

P171 showed highly delayed hair cell degeneration that also shows a correlative loss of sensory innervation in parallel to the base-to-apex progression (Xiang et al., 2003). The surprisingly long retention of hair cells showed at least 6 months. This data suggesting a long-term survival of Pou4f3 null mice beyond some hair cell, also with a delay the innervation retains.

Agreed.

E196 Recent data shows that Sox2 early deletion causes entirely blocks hair cell formation but also found a differential effect of vestibular versus cochlear hair cells (Dvorakova et al., 2019). Given that all cochlear differentiation is absent in these mice it points into a much earlier loss of initial HC induction.

Agreed, we feel that Sox2 plays a very early role in the induction of all hair cells and that Fgf20 and Fgfr3/Prox1 are more likely to be involved in specification of auditory hair cells.

P366 Beyond the general base-to-apex of Atoh1 (Mateo et al., 2005) and apex-to-base Prox1 (Fritsch et al., 2010) is following a counter-differentiation of BDNF (apex-to-base) and Nt-3 (base-to-apex; Farina et al., 2001). Clearly certain projects follow the apex-to-base progression (Chena and Segil, 2002; Kopecky et al., 2011). How is that differentiation progressing from Atoh1 in a counter-direction and why is it in similar NT-3 expression?

We agree that the cochlear is characterized by both apical-to-basal and basal-to-apical gradients. Perhaps the best known of these is the apical-to-basal gradient of terminal mitosis. In some cases, reverse gradients in which one gene is down regulated from base-to-apex while another is upregulated from base-to-apex could represent a replacement of functionality between related genes. So in the situation mentioned, it could be the case that BDNF is down-regulated from base to apex as Nt3 is upregulated to replace the function of BDNF.

--

Reviewer #2 (Remarks to the Author):

Overall comments

- The authors present a cohesive, in-depth analysis of developing mouse cochlear cell transcriptomes. The results confirmed distinctive expression profiles of cells derived from lateral and medial progenitor cells. Additionally, developmental trajectories of hair cells identified genes that may play an important role in cell fate specification and development. Collectively these data provide a valuable resource for characterizing the transcriptional changes that occur during development of the highly distinctive cochlear inner and outer hair cells. This provides a roadmap for eventual identification of genes and regulatory pathways that control inner and outer hair cell specification and development. The manuscript is nicely written.
- The total number of cells included in the study (for all time points?) was 30,000. It is unclear how many cells were collected and analyzed at each time point, and what proportion of these cells were representative of the hair cell (immature or developing) populations. The lack of clarification of the number of cells analyzed at each time point may be construed as misrepresentation of data and should be addressed.

We have added a Table as Supplemental Table 1 that includes details on the total number of cells analyzed at each time point and the total number of each cell type.

- Burns et al. (2015, Nature Communications 6: 8557) published single-cell RNA-seq transcriptomes of cells from neonatal inner ear. Comparison should be made between the two datasets to determine how consistent are the gene expression profiles of different cell types, especially the hair cells.

This is a good point. In order for this comparison to be as meaningful as possible, we considered which data from the two data sets would be the most appropriate to compare. Hair cells from the Burns et al. (2015) study were easily identified based on both expression of a number of hair cell specific genes and on the presence of red fluorescence which was generated through expression of a *Gfi1^{cre}* driver and a tdTomato reporter mouse. For the supporting cells, the most uniform population of cells were in the cluster of cells that showed consistent green fluorescence (labeled as SC in Fig. 7a in Burns et al.). Based on validation of gene expression and the known pattern of expression for the *Lfn3^{Gfp}* line (which is not expressed in inner pillar cells, see Burns et al., Figure 1f and g), this cluster of cells most likely represents outer pillar cells and Deiters' cells. From the new 10X data set, P1 hair cells were already clustered into one group. For the supporting cells, we combined the most relevant supporting cell types, outer pillar cells and Deiters' cells into one group named "support cells". Then we removed all other cell types except for the hair cell and the new "support cell" clusters and then performed differential expression (DE) analysis. From those results we identified the 50 genes most highly differentially expressed in the hair cell or support cell clusters in the 10X P1 data set (these results are provided in a new

Supplemental Table 3). Next, we used the gEAR comparison function to generate a scatter plot of gene expression (average nTPMs per gene per cell) in the hair cell and support cell clusters from Burns et al. We then localized the top 50 DE genes for both the hair cell and support cell clusters from the 10X P1 data set on that graph (new Figure 2m and Supplemental Table 3). The dashed blue line represents equal expression in both cell types. Of the top 50 hair cell DE genes in the 10X data set, 6 were not detected at all in the Burns et al data set. Of the remaining 44 genes, all showed higher levels in hair cells relative to support cells in the Burns et al. data set. Similarly, for the top 50 support cell DE genes, 3 were not detected in the Burns et al., data set, but of the remaining 47 genes, most (at least 40) were more highly expressed in support cells. We have included a description of these results beginning on line 137 and in the Figure 2 legend.

Specific comments

- There are a considerable number of abbreviations used throughout the manuscript, please include a comprehensive list of abbreviations at the end.

A complete list of abbreviations has been added beginning on line 653

- Remove “already” from line 78.

Removed

- Line 130, 159, 180- “see below” is a vague reference. I suggest removing these to avoid confusion.

Removed in all three locations

- After supplemental figures 5 & 6 are referenced in the text, the numbering in the text does not correspond with the numbered figures (missing 7). Please confirm that the numbering of all supplemental figures is correct, both in the text and the attachment.

- Line 241- change to “Fig. 5c”.

We renumbered all the Supplemental Figures

- Line 244- change to “Fig. 5b”.

Changed now on line 271

- Line 275- add “expression of metabolic pathway gene sets”.

Changed as requested now on line 306

- Line 311- change “single cell” to “cell type-specific” to avoid implying that all the other data sets are single cell RNA-seq, as two of them are not.

Changed now on line 343

- While the justification for the discrepancies between datasets is valid (line 322-327), it would be worth noting the number of cells identified as IHCs or OHCs in the dataset (out of the 30,000 cells analyzed) as it may be considerably less than the total number of cells that were analyzed by Liu et al. (2014, J Neurosci 34: 11085-95) and Li et al. (2018, Scientific Data 5:180199), though there is a reference to this in the discussion.

The number of IHCs and OHCs analyzed at P7 has been added to line 338

- Line 412- change “lineal” to “linear”.

This was a typo but not to linear. What we meant to say was that specific supporting cell groups may share lineage relationships with specific hair cell types. So “lineal” has been changed to “lineage” on line 458

- What version of Cell Ranger was used? **Version 2.1.1 now noted on line 506**

- How many total cells were initially sequenced (at each time point)? What was the total number of cells included in the final analysis after applying exclusion parameters (at each time point)?

This data has been added as Supplemental Table 14 and noted in the Methods on line 516.

There are references to cells numbers collected (see Fig. 1b. 14,000 cells at P1), while Fig. 3 does not include the number of HCs included in the analysis. The number of cells collected at E14 and E16 is unclear, while Fig. 7a states that 3,400 P7 cochlear cells were isolated.

We have added a new Supplemental Table 1 that includes data on the total number of cells included at each time point as well as a break down based on cell type. And references to those numbers have been added on lines 67, 151, 200, and 326.

- Line 473- remove extra “(“ at end of sentence.

Removed

- What p-value FDR was used for Monocle analysis?

An FDR of 0.1 was used to select genes that were differentially expressed along the continuum was 10% or .1. We have added this to the Methods section on line 562.

- Line 544 refers to reference 33 as bulk RNA-seq data, but Liu et al. 2014 was microarray data. It should be Li et al (2018) or Liu et al (2018, Front Mol Neurosci 11:356).

We actually changed the way in which this analysis was done. The new description is more accurate about the groups that were used for comparison. The methods used for these comparisons are now described on lines 537-547.

- Line 563- What was the dose of tamoxifen used? Was it an IP injection?

We have added details on the dosage of tamoxifen and the injection protocol beginning on line 630.

- Additionally, the methods should be written to clarify “Pregnant dams were given a single injection of tamoxifen” unless the individual embryos were injected?

The text on lines 635 has been changed to clarify that each pregnant dam was gavaged a total of one time on only one of the three gestational days, E14, E15 or E16.

- Line 571- From what supplier was the SB505124 obtained?

The vendor, SigmaAldrich, has been added on line 644.

- Figures should be examined, and font sizes should be increased when possible.

We have tried to increase font size where possible but did not want to obscure other aspects of the images.

- Figure 1d. Y-axis label? Relative expression quantified as fold change? Is this consistent throughout the other figures? Please clarify.

The Y-axes for all violin plots in the manuscript are normalized, log transformed expression probability values for each gene for all cells within each cluster. An explanation has been added to the legend for Figure 1d and a label has been added to the Y-axis in Figure 1d.

- Figure 3d. The data points for Tmc1 are not clearly visible, consider adding circles or arrows, or changing the color scale.

We added circles to the Tmc1 graph to illustrate that expression is restricted to the most mature hair cells and made a note of this in the legend.

- Figure 3e. What do the red and blue circles denote?

The red circle indicates the HC regulon cluster and the blue circle indicates the SC regulon cluster. A note to that effect has been added to the legend.

- Figure 4c and d. The blue/red/green annotations are confusing. Clarify if the color corresponds to a dot, circle or both by modifying the legend.

The color of the circle in 4c has been changed to black to reduce confusion and the circles in panel 4d have been removed. The legend has been changed on line 766-778 to better explain the annotations.

- Figure 5d and e. Why were two different cell markers used (MYO7A and POU4F3) for the qualitative and quantitative analysis? There should be consistency across both analyses.

The analyses were consistent across all experiments. In all cases, samples were triple-labeled with antibodies against MYO7A, POU4F3 and PROX1. Anti-MYO7A staining fills the hair cell cytoplasm while anti-POU4F3 only labels the hair cell nuclei. Anti-PROX1 staining labels the supporting cell nuclei. To quantify cells in explants, we have developed an automated program for counting nuclei of hair cells and supporting cells labeled with anti-POU4F3 and anti-PROX1, respectively. We have replaced the previous panel 5d with a new image that shows all three makers. We have also included a description of the triple-labeling in the Methods on line 647.

- Figure 5d. Add a scale bar.

We added a scale bar

- Figure 5e. The abbreviations are inconsistent with the text. Change to DIV-2 and DIV-5.

We were actually unclear about what these abbreviations mean. In all cases, explants were maintained for 5 days in vitro. But in the samples listed as DO, this indicates the number of days that the explant was exposed to SB505124. So DO-2 means 2 days with SB505124 followed by three additional days in vitro, but in control media. We have added an explanation of this to the figure legend.

--

Reviewer #3 (Remarks to the Author):

The manuscript from Kolla and Kelly: “Characterization of cochlear hair cell development at the single cell level” provides a very exhaustive characterization of developmental transcriptional profiles, at the single cell level, for the cochlear sensory epithelium.

This is a useful data set for the research community, and will serve as an important reference for future studies. The authors provide developmental trajectories, for different cell populations, including cells in the Kölliker organ, lateral and medial progenitors within the cochlear prosensory domain and inner and outer hair cells.

The large amount of data generated by this approaches and presented here makes however the story in some parts unstructured, and seem to lack focus and depth, in particular for the more mechanistic aspects.

We agree that the large amount of data presented in this manuscript prevents us from doing a more in depth analysis of particular time points or mechanisms. However, we feel that it is important to publish all of this data at one time so that it can be made available to the community. We are in the process of looking at several aspects of this study in greater depth and will publish those results once they are available.

-One of the major findings is the identification, already at E14, of two transcriptionally distinct populations of Sox2 cochlear progenitors, further named lateral and medial progenitors. Analysis at different time points as well as pseudotime analysis seems to confirm that lateral progenitors will give rise either to supporting cells (Deiters) or outer hair cells, while medial progenitors, seem to contribute to inner hair cells and inner border/phalangeal cells. It is not clear however from the trajectory analysis (lines 211-220; figure 4f and supplementary 9 (actually 10), nor legends and methods) if the populations were clustered a priori, before Monocle was used to define trajectories.

The unique cell clusters were identified in a completely unbiased manner using the Seurat “FindClusters” analysis. Subsequent examination of differentially expressed genes in each cluster was used to assign identities. For the Monocle analysis, clusters identities were used to subset the relevant cells. We have added a comment on line 202 to clarify this.

If I understand correctly, the analysis was performed separately on (IHC, MPscs and IPhC) or (LPC, DC, OHC)?.

Yes this is correct

If this is the case, the authors should show what a more unsupervised approach looks like, where medial and lateral progenitors and all deriving cell types are analyzed for the 3 time points.

As requested, we merged prosensory cells, hair cells, Deiters’s cells and inner phalangeal cells from E14, E16 and P1 and performed a trajectory analysis and tSNE plots for the data. The results look very similar to the analysis of lateral cells (Lat. Pro., OHC, DCs)

most likely because of the larger number of these cells in the data set. This result has been added as Supplemental Figure 9 and mentioned in the text beginning on line 240.

Ideally a earlier time point (E13? Should be included).

Unfortunately we do not have cells from the E13 time point and the addition of this data is beyond the scope of this study.

It may be worth to expand figure 4f by combining data from supplementary figure10 (referred to in the text as figure 9).

We have moved the copies of the developmental trajectories that illustrate cellular distribution based on time point to an expanded fig. 4f. The supplemental figure has been adjusted and the legend has been changed.

-Fgfr3 lineage tracing has already been done, giving similar results, using nevertheless later time points (from p1; Cox et al JARO2012) confirming the findings.

We have added a reference to this work on line 227

Others have reported instead defects in outer hair cell formation in FGF20 KO animals (Huh Plos Biology 2012), or lack of all hair cells by blocking antibodies against FGF20 on explant culture. While FGF20 is expressed in medial progenitors, effect on OHC development should be discussed. These findings should be included in the discussion.

The reviewer points out an important omission about the known role of Fgf20/Fgfr1 interactions in specification of the lateral domain of the organ of Corti. We have added a discussion of the relevant findings to the Discussion beginning on line 476.

-The authors identify Tgf β 1 as a putative novel marker of lateral progenitors. The expression of Tgf β 1 does not seem to be exclusively restricted to lateral progenitors (as shown in the tSNE plots Fig 5b), but seems to have higher expression in this population.

We added a comment to note that the scRNAseq data does indicate sporadic expression of both *Tgfr1* and *Fzd9* in cells that are not part of the lateral prosensory clusters on line 262. While not shown in the images in Figure 5c, we do see sporadic smFISH puncta for *Tgfr1* and *Fzd9* outside of the lateral prosensory domain which is consistent with the scRNAseq data. A comment noting this has been added on line 271.

In addition, inhibition of Tgfb signaling using a small molecule inhibits outer hair cell formation in organotypic cultures. The following figure (Figure6) on metabolic changes and associated signature however does not show in any way the contribution of Tgfb signaling into the process, even though it is presented as one of the Tgfb effects. Despite being potentially interesting, a

more detailed mechanistic analysis of the process (with Alk inhibitors or SMAD2 conditional deletions) would be needed to show how much these processes depend on Tgfb signaling. I would suggest removing entirely figure 6.

We agree that the potential role of metabolic pathways in mediating the effects of Tgfb signaling in the cochlea is entirely speculative. However, we feel that it does illustrate one of the potential uses of this data set. In light of the speculative nature of these results, we have reduced the size of the figure and moved it to Supplemental data and have reduced the text related to this result.

-The analysis of deafness-associated genes presented in Figure 2M breaks the story line. In my opinion it could be discussed in conjunction with the same analysis (performed at E14,16 and P7, and presented in supplementary figure 3) at the end of the manuscript.

We agree and have moved this analysis to the end of the manuscript. We have combined the deafness gene panel for P1 with the former supplement panels for E14, E16 and P7 and now have added this as Fig. 7 with a section in the results.

As hair cell development is not the only feature presented, the title could be referring more generally to the development of the cochlear sensory epithelium.

We agree and have changed the title to “Characterization of development of the cochlear epithelium at the single cell level”

Minor comments

Line 28-29 (Abstract)

... comprising several unique cell types... remove “including OHC, DC and PC”, as all are unique cell types.

Removed

Line 111

Tbx expression by smFish :

Tbx2 is indicated as IHC specific and in the text it is mentioned: “it is detected at lower levels in supporting cells”. Expression seems to be very high in the KO region, please rephrase.

Agreed, we have rephased this part of the sentence to include KO on line 116.

OC90 positive population is indicated for different developmental stages but not discussed, it is not clear what it represents-> specify

OC90 cells are positive for Otoconin 90 which is specific for cells in the cochlear roof. We have added a comment about the likely source of these cells and a reference to the Hartman et al paper mentioned below starting on line 82.

OC90 is obviously very highly expressed at mRNA level in the cochlea as shown by several reports. Hartman et al (Frontiers in Neuroscience) reported in situ hybridization of OC90 and showed localization in the roof of the cochlear duct, however here tissue microdissection (even though not described in detail) seems to be excluding this compartment. What are these cells?

As the reviewer surmises, we believe these are cochlear roof cells that were included with our dissociation in an effort to ensure that the cells representing the entire medial to lateral axis of the cochlear floor were represented. This is particularly at the earlier time points where gross morphological landmarks are less clear in microdissection, and is reflected in the number of OC90 cells is highest in the E14 data set and decreases in E16 and then P1. We have added a comment about this beginning on line 83.

Supplementary tables

Excel Tables provided are not labeled as table 1,2,3 etc (as in the text)
This is the kind of file name visible to the reviewer:
224312_0_supp_4078389_pxsd4x_

it is very difficult to find the correct tables the authors refer to→ rename

All tables have been renamed to make them easier to navigate and we have included additional explanations for the content of each table.

Supplementary figures

-Supplementary figure 7 is missing, or not existing and legends are therefore not correct from 7 onwards. In general, given the substantial amount of data presented in the supplementary material, figure legends should be extended, to better understand the content.

Supplementary images have been relabeled and legends have been expanded.

-What is the difference between data presented in supplementary Figure 10a and Figure 5a? From the legend it seems it's the same data. (violin plots for lateral prosensory domain)

Figure 5a shows violin plots for 9 genes that are exclusively expressed in the lateral prosensory population. Supplemental figure 10 (nee 9) now shows dotplots for the top 100 differentially expressed genes in the lateral prosensory cell population. This result includes the 9 genes shown in panel 5a. Dotplots were used to be able to show data on as many genes as possible in relatively limited space. The violin plots provide a different, more detailed way to look at the same data.

-Supplementary figure 6 Tectorin b as highest gene in Lateral progenitors? Is this correct?

Yes that is correct. But Violin plots actually indicate that the average levels of expression based as a probability in a single cell. So, if one were able to randomly select single cells from a bag, on average single lateral prosensory cells would have an expression value for Tectorin b that would be close to 3. All other cells, on average, would have a value of zero. But, this does not mean that no other cells in the cochlea express Tectorin B at E16. And, in fact, the data from Rau et al. (1999) suggests that at least some cells in KO probably also express Tectorin B. But because we pooled all cells except for lateral prosensory cells the relative size of the two cell populations is 527 lateral prosensory cells versus 7434 other cells. In fact, if one generates a violin plot for Tectorin B using all the identified cell types at E16, some cells in the IPC, IPhC, lateral KO, medial prosensory, OHC and Ube2c+ clusters are also positive for Tectorin B (see below). One effect of the pooling method used to identify differentially expressed genes can be a suppression of sporadic expression in other cell clusters.

-Supplementary figure 9 (should be Supplementary figure 8)

We went through and renumbered all the Supplementary Figures.

The text in the manuscript does not reflect what is shown in the figure:

Fgf20 was expressed in MPSCs, inner phalangeal cells and IHCs, while Fgfr3 and Prox1 were expressed in LPSCs, DCs, OPCs and OHCs (Suppl. Fig. 8).

There's no apparent expression of Fgf20 in inner phalangeal cells in the violin plots;

There's no expression of Fgfr3 in OPC, the label indicates inner Pillars.

The reviewer is correct. We changed the text on lines 219 accordingly.

(line 195 in the text, same paragraph, Fgfr3 is labeled as Fgr3)

fixed

Figures

Figure 5 : Orange-shaded area over the smFish prevents seeing the expression of the probe in that area, leave the contour, remove the color.

As requested, we have changed the inner pillar cells to only be outlined.

Methods

The authors indicate that at embryonic time point they have isolated cochleas from 10-12 pups, while for postnatal tissue from 5-8 animals.

It is therefore not clear what the authors refer to with a “minimum of 3 biological replicates was used”. I believe all cochleas from a single developmental stages pooled for analysis?
Please clarify

We have clarified this beginning on line 499. Basically, each biological replicate is a separate isolation of cells from a litter of mice. For embryonic time points, cochlea from 10-12 embryos were required to obtain sufficient cells for a single 10X Chromium run. For post-natal time points, cochlea from only 4-5 animals were sufficient for a single run. So each biological replicate was performed from a separate group of animals on a separate day.

Figure 1. Gene expression along the P1 OHC developmental trajectory. tSNE plots show down regulation of *Atoh1*, *Isl1*, *Prox1* and *Fgfr3* along the OHC tSNE. *Fzd9* shows a similar pattern. To confirm this, we used smFISH to examine *Fzd9* expression at P1 in basal (more mature) and medial (less mature) turns of the cochlea. Results are consistent with the *Fzd9* tSNE plot with more OHC expression of *Fzd9* in the less mature middle turn.

Figure 2. Violin plot for expression of Tectb in single cells at E16. While Tectb is uniformly highly expressed in Lateral prosensory cells, there is sporadic expression in other cell types. These include derivatives of lateral prosensory cells, such as OHCs and IPCs.

Reviewers' Comments:

Reviewer #2:

Remarks to the Author:

The authors present a cohesive, in-depth analysis of developing mouse cochlear cell transcriptomes. The results confirmed distinctive expression profiles of cells derived from lateral and medial progenitor cells. Additionally, developmental trajectories of hair cells identified genes that may play an important role in cell fate specification and development. Collectively, these data provide a valuable resource for characterizing the transcriptional changes that occur during development of the highly distinctive cochlear inner and outer hair cells. The study is appropriate for the journal.

The authors have done a nice job to revise the manuscript. Issues raised by this reviewer have been adequately addressed. However, Fig. 4 C and D still need to be fixed (they said that they did this, but I don't see a difference in the figure). The blue/red/green annotations are confusing in this figure. Clarify if the color corresponds to a dot, circle or both by modifying the legend.

David Z. He

Reviewer #3:

Remarks to the Author:

The manuscript from Kolla and Kelly is improved compared to the previous version. Most of my comments have been properly addressed. I have one remaining point and some additional minor comments:

> The mechanistic data on TGFb is very superficial. The authors show the effect of antagonizing TGFb signaling with SB505124 using E14.5 explants (Fig 5d-e) and suggest that inhibition of TGFb signaling halts differentiation, and does not induce cell death, as after removal of the compound hair cell (OHC) counts are normalized.

"We observed a significant loss of OHCs, but no change in PROX1+ SCs, in response to inhibition of Tgfβr1 (Fig. 5d). To determine whether inhibition of Tgfβr1 leads to cell death or an inhibition of cellular differentiation, explants were treated with 20 μM SB505126 for 2 DIV, followed by an additional 3 DIV in control media. Results indicated a recovery of OHC formation (Fig. 5e)"
(should be SB505124)

The authors talk about "differentiation", but it is not clear to what exactly they refer to:

- a) maturation, of a already committed HC with increase expression of MYO7a and POU4f3?
- b) specification from lateral progenitors?

In the second case, one would expect that inhibition of differentiation from LPC would be matched by a higher number PROX1+ cells. The authors indicate there's instead no change in Prox1 cells, and in the new panels provided (PROX1 staining) it almost looks like also these cells have decreased.

Has this been quantified?

How is differentiation recovered if the Prox1 cell number does not change? Proliferation and subsequent differentiation of Prox1 cells?

The authors should clarify this point.

Minor comments

1) Description of figure2 in the text (line 115-126):

Not all the markers presented in the figures are mentioned in the text (example Ccer2, sox2,

p27)), and the order of the panels is not matching the order of the text.
It would be easier to read and find the panels if these were mentioned in order of appearance.

2) There's no method concerning smFISH. This It should be added.

3) Figure 2: expression of TBX2,

The authors have changed the text concerning it's expression in the KO, but in the figure legends it remains IHC only.

"Tbx2 , which was found only in the IHC cluster, is localized to IHCs with weak expression in surrounding cells" modify

4) Figure3b shows the expression of IHC and OHC genes. Also in this case, the order in which the genes are ranked and the text that describes the figures could be matched. (OHC specific genes are difficult to find as not clustered together)

5) Line 226: Monocle analysis of LPsCs, OHCs and DCs failed to produce a bifurcated trajectory (Fig. 226 4fSuppl. Fig. 8).

Suppl Fig 8 shows Medial progenitors only.

6) Concerning the TCA/glutamine metabolism data set, I believe the manuscript reads better now with the data in the supplementary material. As I mentioned before, I would rather exclude it. I leave the decision to the editor.

Kolla et al. NCOMMS 19-31026

Reviewers' comments:

Reviewer #2 (Remarks to the Author):

The authors present a cohesive, in-depth analysis of developing mouse cochlear cell transcriptomes. The results confirmed distinctive expression profiles of cells derived from lateral and medial progenitor cells. Additionally, developmental trajectories of hair cells identified genes that may play an important role in cell fate specification and development. Collectively, these data provide a valuable resource for characterizing the transcriptional changes that occur during development of the highly distinctive cochlear inner and outer hair cells. The study is appropriate for the journal.

The authors have done a nice job to revise the manuscript. Issues raised by this reviewer have been adequately addressed. However, Fig. 4 C and D still need to be fixed (they said that they did this, but I don't see a difference in the figure). The blue/red/green annotations are confusing in this figure. Clarify if the color corresponds to a dot, circle or both by modifying the legend.

We have changed the wording in the legend, line 740-742 to more clearly describe the colors in figure 4c and 4d.

David Z. He

Reviewer #3 (Remarks to the Author):

The manuscript from Kolla and Kelly is improved compared to the previous version. Most of my comments have been properly addressed. I have one remaining point and some additional minor comments:

> The mechanistic data on TGF β is very superficial. The authors show the effect of antagonizing TGF β signaling with SB505124 using E14.5 explants (Fig 5d-e) and suggest that inhibition of TGF β signaling halts differentiation, and does not induce cell death, as after removal of the compound hair cell (OHC) counts are normalized.

“We observed a significant loss of OHCs, but no change in PROX1+ SCs, in response to inhibition of Tgf β 1 (Fig. 5d). To determine whether inhibition of

Tgfb β 1 leads to cell death or an inhibition of cellular differentiation, explants were treated with 20 μ M SB505126 for 2 DIV, followed by an additional 3 DIV in control media. Results indicated a recovery of OHC formation (Fig. 5e)”
(should be SB505124)

The authors talk about “differentiation”, but it is not clear to what exactly they refer to:

- a) maturation, of a already committed HC with increase expression of MYO7a and POU4f3? or
- b) specification from lateral progenitors?

In the second case, one would expect that inhibition of differentiation from LPC would be matched by a higher number PROX1+ cells. The authors indicate there’s instead no change in Prox1 cells, and in the new panels provided (PROX1 staining) it almost looks like also these cells have decreased.

This is a good point. As the reviewer points out, the most likely explanation is option a above, that developing OHCs are halted at a stage after down regulation of Prox1 but prior to ongoing maturation. We have changed the text in lines 284-285 and line 469 to reflect this more specific result/conclusion.

Has this been quantified?

Yes, there was not a significant change in Prox1 positive cells which was already noted on line 280.

How is differentiation recovered if the Prox1 cell number does not change?
Proliferation and subsequent differentiation of Prox1 cells?
The authors should clarify this point.

We did not see any indication of proliferation as removal of SB505124 did not lead to an increase in either OHCs or Prox1+ cells as would be expected if there was renewed proliferation. So, as discussed above, the most likely explanation is an arrest in hair cell development after down-regulation of Prox1 which we have addressed.

Minor comments

- 1) Description of figure2 in the text (line 115-126):

Not all the markers presented in the figures are mentioned in the text (example Ccer2, sox2, p27)), and the order of the panels is not matching the order of the text.

It would be easier to read and find the panels if these were mentioned in order of appearance.

We have changed the text to mention all the genes depicted and to describe the results in an order that matches the order of the panels in figure 2.

2) There's no method concerning smFISH. This It should be added.

We have added methods for smFISH beginning on line 606.

3) Figure 2: expression of TBX2,

The authors have changed the text concerning it's expression in the KO, but in the figure legends it remains IHC only.

"Tbx2 , which was found only in the IHC cluster, is localized to IHCs with weak expression in surrounding cells" modify

We have modified lines 708 and 709 to indicate expression of Tbx2 in IHCs and KO.

4) Figure3b shows the expression of IHC and OHC genes. Also in this case, the order in which the genes are ranked and the text that describes the figures could be matched. (OHC specific genes are difficult to find as not clustered together)

The text was changed on lines 155- 158 to indicate the full list, in order, of genes expressed in IHCs and OHCs. Similar changes were made in the figure legend, lines 732-734, to also list the same genes in order.

5) Line 226: Monocle analysis of LPsCs, OHCs and DCs failed to produce a bifurcated trajectory (Fig. 226 4fSuppl. Fig. 8).
Suppl Fig 8 shows Medial progenitors only.

The lateral prosensory trajectory was moved to figure 4f. the reference to suppl. Figure 8 has been removed.

6) Concerning the TCA/glutamine metabolism data set, I believe the manuscript reads better now with the data in the supplementary material. As I mentioned before, I would rather exclude it. I leave the decision to the editor.

**We appreciate the reviewers willingness to defer this decision to the editor.
We would like to keep this work in the manuscript as we feel it will be
useful to the community.**